# Pruning neural network models for gene regulatory dynamics using data and domain knowledge

**Intekhab Hossain**[*, §]
Department of Biostatistics
Harvard T.H. Chan School of Public Health
Boston, MA 02115
ihossain@g.harvard.edu

**Jonas Fischer**[*]
Dep. for Computer Vision and Machine Learning
Max Planck Institute for Informatics
Saarbrücken, Germany
jonas.fischer@mpi-inf.mpg.de

**Rebekka Burkholz**[†]
Helmholtz Center CISPA
for Information Security
Saarbrücken, Germany
burkholz@cispa.de

**John Quackenbush**[†]
Department of Biostatistics
Harvard T.H. Chan School of Public Health
Boston, MA 02115
johnq@hsph.harvard.edu

## Abstract

The practical utility of machine learning models in the sciences often hinges on their interpretability. It is common to assess a model's merit for scientific discovery, and thus novel insights, by how well it aligns with already available domain knowledge–a dimension that is currently largely disregarded in the comparison of neural network models. While pruning can simplify deep neural network architectures and excels in identifying sparse models, as we show in the context of gene regulatory network inference, state-of-the-art techniques struggle with biologically meaningful structure learning. To address this issue, we propose DASH[‡], a generalizable framework that guides network pruning by using domain-specific structural information in model fitting and leads to sparser, better interpretable models that are more robust to noise. Using both synthetic data with ground truth information, as well as real-world gene expression data, we show that DASH, using knowledge about gene interaction partners within the putative regulatory network, outperforms general pruning methods by a large margin and yields deeper insights into the biological systems being studied.

## 1 Introduction

With ever-growing neural network architectures encouraged by the success of overparametrization, with over a trillion parameters in a single model such as GPT4, there is a similarly growing demand for sparser, more parameter-efficient neural networks that are more resource-friendly and interpretable. The Lottery Ticket Hypothesis (LTH) provides an empirical existence proof of sparse, trainable network architectures [18], that eventually achieve a similar performance as their dense counterpart. Subsequent work introduced *structured* pruning approaches, facilitating group-wise neuron- [32, 62], or channel-sparsity [34, 25, 26], which are, however, focused on the structure of the *architecture design*, aiming for better alignment with hardware implementations to eliminate operations, rather than structure that reflects *relevant domain information*. Especially for scientific discovery, an

---

[‡]Code is available at https://github.com/QuackenbushLab/DASH.

[*,†] These authors contributed equally to this work.

[§] IH is currently employed by Analysis Group, Inc.

38th Conference on Neural Information Processing Systems (NeurIPS 2024).

alignment of learned structure with such domain knowledge is, however, essential for interpretability, as only then the model represents meaningful domain-relevant relations. Such problem settings often occur for example in physics or biology where a learned model should give an explanation to be able to form a hypothesis. This poses the question: How can we select among multiple predictive models and promote the search for *meaningful* neural network structures?

To guide the learning process, we argue that we need additional problem-specific structural information, and should leverage any available - and reliable - domain knowledge. One of the fundamental tasks of molecular biology is to understand the *gene regulatory dynamics* in health and disease. Gene regulatory dynamics describe the changes of the expression of a gene—the generation of small copies of a DNA segment that can serve among others as blueprint for proteins—dependent on other regulatory factors such as transcription factors, which are proteins that bind next to the gene (DNA segment) to modulate its expression. Yet, the exact dynamics are far from understood and changes in these dynamics can be drivers for diseases such as cancer. As such, improving an understanding of the mechanics behind these dynamics increases the understanding of the disease and can ultimately inform therapy design. This problem setting of estimating gene regulatory dynamics requires high interpretability, as an understanding of the true biological mechanics—the relationship between regulatory factors and a gene's expression—is needed, as well as sample-efficiency, as generating time-course data even for a few patients is extremely expensive. While models to estimate gene regulatory dynamics have been suggested [14, 1, 60, 27], none of these is particularly sparse or interpretable. We propose a new approach of network sparsification that *guides pruning by domain knowledge* implemented for a neural model for estimating gene regulatory dynamics, which yields networks that are very sparse, align with underlying biology, while accurately predicting dynamics.

This idea of *prior-informed* or domain-aware pruning is at the heart of this paper. In particular, we propose DASH (**D**omain-**A**ware **S**parsity **H**euristic, Fig. 1), an iterative pruning approach that scores parameters taking into account structural domain-knowledge. With DASH, it is possible to control the level of prior information taken into account for pruning and it *automatically finds an optimal sparsity level* aligned with both the prior and the data. Considering the task of estimating gene regulatory dynamics, we first show in synthetic experiments that DASH generally outperforms standard (task- and architecture agnostic) pruning as well as task-specific pruning approaches. On real data with a reference gene regulatory network (GRN) derived from gold standard biological experiments, we show that DASH better recovers the reference GRN and reflects more biologically plausible information. On recent single cell data on blood differentiation, we show that DASH, in contrast to existing work, identifies biologically relevant pathways that can be used to generate new insights and inform domain experts. We anticipate that our work serves both for future benchmarking on how well pruning approaches are in structure learning, as well as a blueprint for guided network pruning in fields where domain knowledge is readily available, such as in other hard sciences including physics or material science, where knowledge about variables, e.g., associations between atoms or molecules or equations relating quantities in a system, is available.

## 2   Related work

The Lottery Ticket Hypothesis (LTH) [18] provides an empirical existence proof of sparse, trainable neural network architectures. It conjectures that dense, randomly initialized neural networks contain subnetworks that can be trained in isolation with the same training algorithm that is successful for the dense networks. However, a strong version of this hypothesis [51, 65], which has also been proven theoretically [42, 48, 46, 17, 5, 12, 15], suggests that the identified initial parameters are not only specific to the sparse structure but also the learning task and benefit from information about the larger dense network that has been pruned [47]. Acknowledging this strong relation, other works have proposed to combine mask and parameter learning directly in continuous sparsification approaches [53] that employ regularization strategies that approximate L0 penalties [53, 40, 31, 55]. In the following, we recap the key ideas behind the methods most relevant to ours.

**Explicit pruning-based approaches**

**Magnitude pruning (MP)** In magnitude pruning (MP) a neural network is trained and then (post-hoc) pruned to a desired sparsity level by masking the corresponding proportion of *smallest magnitude* weights. This smaller, masked network is then further trained, reminiscent of fine-tuning [22].

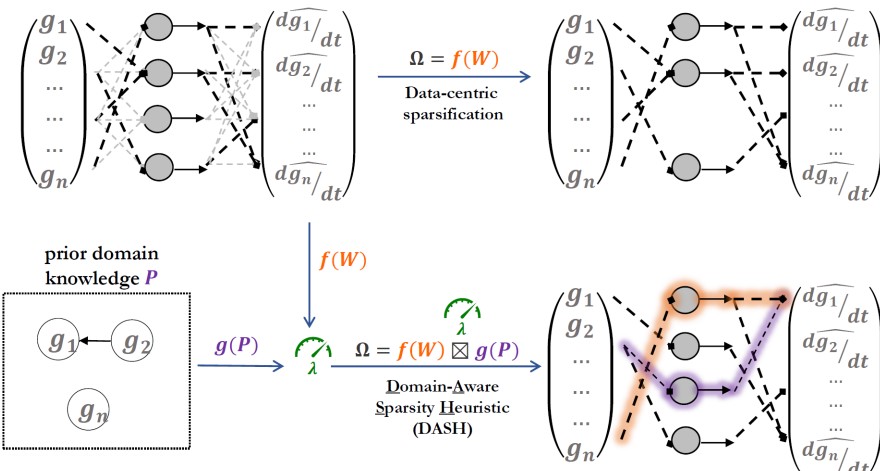

Figure 1: *DASH.* A NN, here a neural ODE for gene regulatory dynamics, is traditionally sparsified in a data-centric way (top). Pruning is done based on data alone, the pruning score $\Omega$ is a function of the learned weights $W$. Such sparsified models often do not learn plausible relationships in the data domain. We propose DASH (bottom), which additionally incorporates domain knowledge $P$ into the pruning score $\Omega$, yielding sparse networks giving meaningful and useful insights into the domain.

More formally, we start with a fully connected neural network $\text{NN}_\Theta$ with $L$ layers parametrized by $\Theta := \{(W^l, b^l)\}_{l=1}^L$. MP is performed after training is complete (i.e. post hoc). For a suitable threshold of $p\%$, MP "prunes" the trained $\Theta$ by setting the smallest (absolute value) $p\%$ of weights in $\Theta$ to 0. The choice of parameters to prune can either be made in an unstructured way, by choosing the the lowest $p\%$ across all $\{W^l\}_{l=1}^L$, semi-structured, by pruning the lowest weights per layer, or in a structured way such that, e.g., neurons with lowest $p\%$ average outgoing weight are pruned. Once pruning is complete the $p\%$-sparse $\text{NN}_{\Theta'}$ is fine-tuned on the data so that $\Theta'$ is learned appropriately.

**Iterative magnitude pruning (IMP)** [18] suggest to alternate between training and magnitude pruning, iteratively sparsifying the network, to date still of the most successful pruning strategies. In practice, a pruning schedule is used to *iteratively* sparsify $\Theta$, until a $\Theta'$ with desired target sparsity or predictive performance plateau is reached. Importantly, weights are reset to initial pre-training values, either after each round of pruning or once after the target sparsity has been reached.

**Pruning with rewinding** [52] have demonstrated that rewinding weights to an earlier training point—a compromise between fine-tuning of MP and reset to initialization of IMP— provides good performance, which also has been suggested in the context of Neural ODEs as SparseFlow [37].

**Implicit penalty-based approaches**

**C-NODE** [2] seek to reduce the overall number of input-output dependencies (i.e. paths of contribution from input neuron $i$ to output neuron $j$) through $\{W^l\}_{l=1}^L$. The approach can result in both feature and weight sparsity in NeuralODEs.

**$L_0$** [40] incorporate a differentiable $L_0$ norm regularizer term in the objective. It implicitly prunes the network by encouraging weights to get exactly to zero. The $L_0$ regularizer is operationalized using non-negative stochastic gates which act as masks on the weights.

**PathReg** [1] innovatively combines the strengths of both C-NODE and the $L_0$ approach to promote both weight and feature sparsity in NeuralODEs. It uses stochastic gates similar to [40] and add a C-NODE-inspired penalty terms that constrain the overall number of input-output paths by regularizing the *probability* of any path from input $i$ to output $j$ being non-zero.

**Modeling gene regulatory dynamics**   As application, we consider estimation of gene regulatory dynamics. Early work, such as COPASI [44] use a fully parametric modeling approach that are limited in their prediction capabilities. With recent advances in machine learning, tools such as Dynamo [50], PROB [57], and RNA-ODE [39] aim to learn regulatory ODEs using sparse kernel regression, Bayesian Lasso, and random forests, respectively. Leveraging high flexibility and performance of neural models, PRESCIENT [64] uses a simple NN to learn regulatory ODEs, whereas tools such

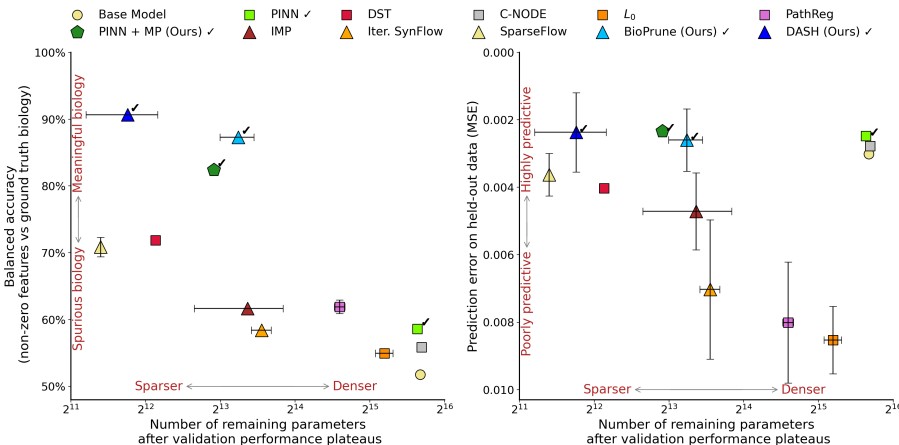

Figure 2: *Results on simulated data.* We visualize performance of pruning strategies in comparison to original PHOENIX (baseline) in terms of achieved sparsity (x-axis) and balanced accuracy (y-axis) of the recovered gene regulatory network against the ground truth on the SIM350 data with 5% noise. Error bars are omitted when error is smaller than depicted symbol. ✓ indicate methods that leverage prior information. Top left is best: recovering true, inherently sparse biological relationships.

as DeepVelo [9] and sctour [35] have a variational autoencoder as backbone. The latest line of research [14, 1, 60, 27] uses neural ordinary differential equations [8]. However, a key limitation of these methods is the lack of interpretability arising from non-sparse dynamics that do not align with ground truth biology. Consequently, the induction of sparsity in gene regulatory ODEs has been an active area of research with C-NODE and PathReg as most recent advances[2, 1], the achieved sparsity levels are, however, not yet sufficient to capture the relevant biology.

## 3   Domain-aware pruning with DASH

While the above sparsification strategies have shown to perform well in various settings, the resulting models are often either not particularly sparse or do not reflect meaningful domain knowledge. We hypothesize that this is due to two reasons: (1) the difficulty of identifying a good sparse network and (2) the current focus on *hardware-centric* rather than *task-centric* pruning, valuing structural pruning of a model in terms of groups of neurons (layers, channels) over structural pruning reflecting task-specific knowledge. To overcome these problems, we suggest to ground the model search (here, the network training) with existing domain-specific knowledge, which eases identifiability due to the introduced constraints and enables task-aware pruning to identify meaningful domain knowledge.

In the following, we propose DASH (**D**omain-**A**ware **S**parsity **H**euristic), an iterative pruning-based approach that accounts for prior knowledge by scoring parameters in terms of their alignment with this prior, and show its usefulness for a neural model for the inference of gene regulatory dynamics. We assume the domain knowledge to be given as input-output relations (e.g., known protein—gene interactions in molecular biology), for which we want the network flow between any input and output to align with. Suppose our domain knowledge for a task from this domain is given as a real-valued relationship graph $G = (V, E)$, where nodes are relevant entities from the domain, e.g. genes or proteins, and edges are a strength of association between these entities, e.g. evidence of association derived from literature or experiments. Examples for such relational information in case of protein—gene associations can be derived from protein binding profiles [56]. For the rest of the paper, we will assume that this domain knowledge is given as a matrix $\boldsymbol{P} \in \mathbb{R}^{k \times r}$, which encodes the strength of association between the $k$ inputs and $r$ outputs for our task of interest, such as known proxies of protein—gene interactions . Intuitively, for a one-layer neural network, we encourage pruning scores for a (neural) network edge to be proportional to the corresponding edge in the prior knowledge graph $G$ while still taking into account the data-specific knowledge, thus enabling learning

of new knowledge and robustness to wrong or missing information in the prior. We begin with this simple base case of task-aware pruning of a fully connected neural network with $L = 1$ layer and extend to more layers below.

**DASH for $L = 1$.** For a single layer NN, with $k$ input and $r$ output neurons and corresponding weight matrix $\boldsymbol{W} \in \mathbb{R}^{r \times k}$, we compute non-negative **pruning scores $\boldsymbol{\Omega} \in \mathbb{R}^{r \times k}$** by leveraging the **domain knowledge $\boldsymbol{P} \in \mathbb{R}^{r \times k}$**. In practice, we allow balancing between *data-driven* and *prior-knowledge-driven* pruning, implemented through a convex combination of the learned weights $\boldsymbol{W}$ and prior domain knowledge $\boldsymbol{P}$ controlled by the parameter $\lambda \in [0, 1]$. Alternating between training and pruning akin to Iterative Magnitude Pruning [18], we set the following pruning score during a pruning phase:

$$\boldsymbol{\Omega}^{(t)} := (1 - \lambda)\widetilde{|\boldsymbol{W}^{(t)}|} + \lambda|\boldsymbol{P}| \,,$$

where $\widetilde{|\boldsymbol{W}^{(t)}|}$ represents the appropriately normalized matrix (details in Appendix B.2) of absolute weights as learned up to epoch $t$. We then prune the parameters in $\boldsymbol{W}^{(t)}$ corresponding to the lowest absolute $p_t\%$ of entries in $\boldsymbol{\Omega}^{(t)}$, where $p_t$ is the desired sparsity level at time $t$ given by a schedule.

**DASH for $L = 2$.** For two-layered NNs with weights $\boldsymbol{W_1} \in \mathbb{R}^{m \times k}, \boldsymbol{W_2} \in \mathbb{R}^{r \times m}$, i.e $k$ inputs, $r$ outputs, and $m$ hidden neurons, we consider knowledge about input-output relationships $\boldsymbol{P} \in \mathbb{R}^{r \times k}$ as before. We can additionally use further knowledge about input-input relationships $\boldsymbol{C} \in \mathbb{R}^{k \times k}$. In molecular biology this could be information about binding or interaction partners available in databases such as STRINGDB [58], which has also been employed to guide static gene regulatory network inference [61], or co-regulators, derived from co-occurrence of proteins. Intuitively, we now project pruning scores for the first layer to the prior knowledge about input-input relations, encouraging closeness to this prior, while projecting the product of pruning scores of first and second layer to known input-output relations, reflecting the flow of information from input to output through these two layers. Given that $\boldsymbol{W_1^{(t)}}$ represents how the $k$ inputs are encoded by $m$ neurons, and $\boldsymbol{\Omega_1^{(t)}}$ are the corresponding pruning scores, we surmise that the matrix product $\boldsymbol{\Omega_1^{(t)}}^\intercal \boldsymbol{\Omega_1^{(t)}} \in \mathbb{R}^{k \times k}$ should approximately align with the prior knowledge $\boldsymbol{C}$. Since solving $\boldsymbol{\Omega_1^{(t)}}^\intercal \boldsymbol{\Omega_1^{(t)}} = C$ is not directly feasible we initialize $\boldsymbol{\Omega_1^{(0)}}$ randomly, and resort to solving a recurrence relation version of problem, that is $\boldsymbol{\Omega_1^{(t-1)}}^\intercal \boldsymbol{\Omega_1^{(t)}} = C$. Using the left and right pseudo-inverse to obtain a solution to the above, defined as $\mathtt{PInv}_L(X) = (X^\intercal X)^{-1} X^\intercal$ and $\mathtt{PInv}_R(X) = X^\intercal (X X^\intercal)^{-1}$ respectively:

$$\boldsymbol{\Omega_1^{(t)}} := (1 - \lambda_1)\widetilde{|\boldsymbol{W_1^{(t)}}|} + \lambda_1 \left| \mathtt{PInv}_L\left(\boldsymbol{\Omega_1^{(t-1)}}^\intercal\right) \cdot \boldsymbol{C} \right|.$$

$\mathtt{PInv}_L\left(\boldsymbol{\Omega_1^{(t-1)}}\right) \cdot \boldsymbol{C}$ encourages $\boldsymbol{\Omega_1^{(t)}}^\intercal \boldsymbol{\Omega_1^{(t)}}$ to iteratively align with $\boldsymbol{C}$ as $t$ increases (i.e. as training progresses). With $\boldsymbol{\Omega_1^{(t)}}$ fixed, we can update scores $\boldsymbol{\Omega_2^{(t)}}$ of the second layer parameters $\boldsymbol{W_2^{(t)}}$. Since the product $\boldsymbol{W_2^{(t)}} \cdot \boldsymbol{W_1^{(t)}} \in \mathbb{R}^{r \times k}$ represents the overall flow of information from inputs to outputs at epoch $t$, we surmise that $(\boldsymbol{\Omega_2^{(t)}} \boldsymbol{\Omega_1^{(t)}}) \in \mathbb{R}^{r \times k}$ should reflect $\boldsymbol{P}$. We thus get

$$\boldsymbol{\Omega_2^{(t)}} := (1 - \lambda_2)\widetilde{|\boldsymbol{W_2^{(t)}}|} + \lambda_2 \left| \boldsymbol{P} \cdot \mathtt{PInv}_R\left(\boldsymbol{\Omega_1^{(t)}}\right) \right|.$$

Similar to the case for DASH for $L = 1$ layer, we can now prune the parameters of $\boldsymbol{W_1^{(t)}}$ and $\boldsymbol{W_2^{(t)}}$ based on the magnitude of pruning scores $\boldsymbol{\Omega_1^{(t)}}$ and $\boldsymbol{\Omega_2^{(t)}}$, respectively.

**DASH for $L > 2$** For many interpretability-centric tasks, including our application to gene regulatory networks, small architectures of $L = 2$ are common, as domain experts are interested in understanding the exact flow of information through the network. Furthermore, we know that two-layer neural networks exhibit universal approximation [11]. We however hypothesize that the technique of computing pruning scores by fixing those of preceding layers can be extended to a larger number $L$ of fully connected layers and elaborate in App. B.6.

**Flexibility** While the $\lambda_l$ can be tuned using cross-validation (see App. B.4), we note that it allows for flexibly encoding different pruning philosophies. Specifically, when $\lambda_l = 0 \; \forall l$, DASH corresponds to SparseFlow, and when $\lambda_l = 1 \; \forall l$, DASH represents fully prior-based sparsification (which we term "BioPrune" and consider as experimental baseline).

Table 1: *Synthetic data results.* We give model sparsity, balanced accuracy with respect to edges in the ground truth gene regulatory network, mean squared error of predicted gene regulatory dynamics on the test set, and number of epochs (till validation performance plateaus) as proxy of runtime. ✓ is used to indicate methods that leverage prior information. Results are on SIM350 data with 5% noise.

| Strategy (✓ = prior-informed) | | Sparsity(%) | Bal. Acc.(%) | MSE ($10^{-3}$) |
|---|---|---|---|---|
| None/Baseline [27] | | $7.5 \pm 0.1$ | $51.8 \pm 0.03$ | $3.0 \pm 0.4$ |
| Penalty-based (implicit) | $L_0$ [40] | $33.8 \pm 4.7$ | $55.0 \pm 0.5$ | $8.5 \pm 1.0$ |
| | C-NODE [2] | $6.2 \pm 0.5$ | $55.9 \pm 0.1$ | $2.8 \pm 0.6$ |
| | PathReg [1] | $56.5 \pm 1.5$ | $61.9 \pm 1.0$ | $8.0 \pm 1.8$ |
| | PINN [27] ✓ | $9.9 \pm 0.4$ | $58.6 \pm 0.7$ | $2.5 \pm 0.2$ |
| | DST[38] | $92.8 \pm 0.3$ | $71.9 \pm 0.5$ | $4.0 \pm 0.5$ |
| Pruning-based (explicit) | IMP [18] | $81.9 \pm 6.6$ | $61.7 \pm 0.7$ | $4.7 \pm 1.1$ |
| | Iter. SynFlow [59] | $79.3 \pm 1.2$ | $58.4 \pm 0.6$ | $7.0 \pm 2.1$ |
| | SparseFlow [37] | $\mathbf{96.0 \pm 0.01}$ | $70.9 \pm 1.5$ | $3.6 \pm 0.6$ |
| | BioPrune (Ours, see 3) ✓ | $83.5 \pm 1.9$ | $87.3 \pm 0.8$ | $2.6 \pm 0.9$ |
| | DASH (Ours) ✓ | $94.6 \pm 1.2$ | $\mathbf{90.7 \pm 0.4}$ | $2.4 \pm 1.2$ |
| Hybrid | PINN + MP (Ours) ✓ | $87.0 \pm 0.01$ | $82.4 \pm 0.2$ | $\mathbf{2.3 \pm 0.3}$ |

# 4 Task-aware pruning for sparse gene regulatory dynamics

Perhaps one of the most interesting applications of Machine Learning is in the field of Molecular Biology with the goal of understanding human health and disease. A central mechanisms in humans is the process of gene expression in each cell. There, copies of short segments of our genome are produced. These copies are among other things the blueprint for the production of different proteins, which are needed virtually everywhere in our bodies. If this tightly regulated process of gene expression goes wrong, for example because of a mutation in our genome, this can have profoundly bad effects, such as in the case of cancer. As such, studying this process is of great interest to understand and improve human health and discover new therapeutic targets.

Here, we consider the task of predicting the regulatory dynamics of gene expression. To be able to understand the model and transfer it to clinical practice, interpretability is key. The most recent developments in modeling gene regulatory systems allow to model actual (temporal) regulatory dynamics, but require complex models, such as NeuralODEs, that hinder interpretability. While state-of-the-art results are now achieved with shallow architectures [27] that are more tractable than deep, heavily over-parameterized networks, these models still encode information across many thousands of weights and we show experimentally that such information does not reflect true biology well. In fact, true gene regulatory networks and hence their underlying dynamics are inherently sparse [6]. This sparsity should be properly reflected by neural dynamics models. The PHOENIX NeuralODE model will serve as our base model for applying sparsification strategies and we show that pruning aligned with prior domain knowledge improves interpretability as well as quality of inferred (new) knowledge.

In a nutshell, given a time series gene expression sample for $k$ genes, PHOENIX uses NeuralODEs to construct the predicted trajectory between gene expression $\boldsymbol{g}(t) \in \mathbb{R}^k$ (inputs) at time $t = t_i$ to any future expression $\widehat{\boldsymbol{g}}(t_{i+1})$ (outputs), by implicitly modeling the RNA velocity ($d\boldsymbol{g}/dt$). PHOENIX uses biokinetics-inspired activation functions to separately model additive and multiplicative co-regulatory effects. The trained model encodes the ODEs governing the dynamics of gene expression, which can be directly extracted for biological insights. We apply DASH to PHOENIX and give a brief review of the PHOENIX architecture in App. B.10 and a detailed account on how to apply DASH to this architecture in App. B.11. Next, we provide experiments on synthetic and real data showing the advantages of prior-informed pruning on the task of predicting gene-regulatory dynamics.

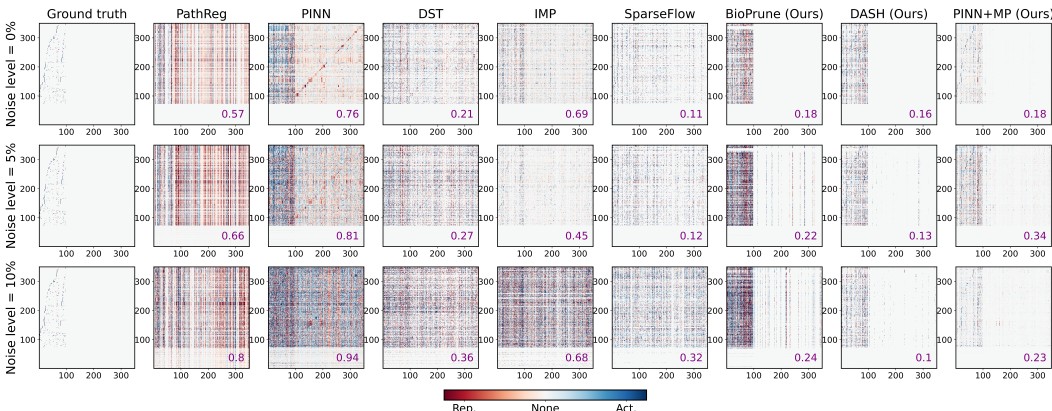

Figure 3: *Reconstruction of ground truth relationships.* Estimated effect of gene $g_j$ (x-axis) on the dynamics of gene $g_i$ (y-axis) in SIM350 for different levels of noise (rows). Ground truth is given on the left, our suggested approach and baselines (DASH, BioPrune, and PINN+MP) on the right with mean squared error between inferred regulatory relationships and ground truth in purple.

## 5 Experiments

For evaluation we consider synthetic data from an established simulator tool [4], as well as real world data of gene expression from breast cancer tissue [13], from yeast with synchronized cell cycle [49], and from human bone marrow [1]. In case of synthetic data, we use the ground truth regulatory system from the generating model for validation. For breast cancer and yest cell cycle data we use additional experimental data (ChIP-seq) from the corresponding studies, which are independent gold-standard biological experiment measuring sample-specific TF–gene interactions, to evaluate inferred regulatory relationships. We measure the correctness of a GRN learned by a model (see App. B.10.4) in terms of balanced accuracy, which measures whether an edge is correctly reconstructed weighted by the sparsity of the aforementioned ground truth graph. To evaluate predictive performance for real data, we use a 6% hold-out test set for breast cancer and one of the biological replicates hold out from training for the yeast data. As prior knowledge we leverage general information of transcription factor binding to gene promoter regions as prior information, which can be computed from binding motif matches with the corresponding genome (human respectively yeast). The result is a matching score that can be thresholded to get a $0, 1$-based matrix encoding which (TF-encoding) gene has a relationship with which other gene. We follow the approach of Guebila et al. [3] to obtain matrix $P$. As prior $C$, we use the STRING database [58], which gives a general (i.e., not tissue-specific) graph of protein-protein interaction. Here, we use the interactions based on experimental evidence only and employ a cutoff of .6 to get a binary adjacency matrix. (for more details, see App. B.3).

To compare pruning strategies, we consider the PHOENIX model as a basis, which is the state-of-the-art NeuralODE for estimating gene regulatory dynamics [27] and provide an ablation on a standard MLP architecture (see App. Tab. 7, App. Tab. 10, and App B.12). We compare DASH against the PHOENIX model without additional pruning as performance reference, and suggest two simple yet powerful baselines, which is post-hoc magnitude pruning of weights followed by finetuning (PINN+MP), and BioPrune, a fully prior-based pruning (cf. Sec. 3). From the literature, we consider $L_0$-regularized pruning [40], C-NODE [2], and PathReg [1], which have been recently proposed for the inference of sparse gene-regulatory relationships, PHOENIX with biological regularization [27], and dynamic sparse training (DST) [38], all of which are implicit pruning approaches. We further consider explicit, iterative score-based pruning approaches including Iterative Magnitude Pruning (IMP) [18], the flow-based model-agnostic pruning method SynFlow [59], and the flow-based Neural ODE pruning SparseFlow [37]. We tune hyperparameters, including $\lambda$ for DASH, on a validation set. Unlike other methods (e.g., $L_0$) DASH does not prolong the runtime of training much. A common pruning schedule where pruning scores are computed once every 10 epochs only increases runtime by <2% for the full model fitting process.

**Simulated gene regulatory systems** We simulate gene expression time-series data from a fixed dynamical system, the ground truth was thus known (see App. B.1). In short, we adapt `SimulatorGRN` [4] to generate noisy time-series expression data from two synthetic gene regulatory systems (SIM350 and SIM690, consisting of 350 and 690 genes, respectively). We split trajectories into training (88%), validation (6% for tuning $\lambda$), and testing (6%).We evaluate all methods in terms of achieved sparsity and MSE of predicted gene expression values on the test set (App. B.2, B.4) and investigate biological plausibility by calculating balanced accuracy of regulatory relationships extracted from the PHOENIX model (for details, see App. B.5, B.10.4), We here report the results for the data of 350 genes and 5% noise, noting that results are consistent across different noise levels and with more number of genes (see App. Sec. A.1).

A general trend across all experiments that aligns with our initial motivation is that dense models (sparsity $< 50\%$) have a significantly worse reconstruction of the underlying biology – the ground truth GRN – than sparse models (sparsity $> 80\%$) (see Fig. 2). Furthermore, we see that DASH retrieves not only among the sparsest networks, but also reflects the underlying GRN best across all methods, outperforming comparably sparse IMP by about 20 percentage points accuracy in different settings, even with decrease in quality of the prior (see sensitivity analysis in App. Tab. 6). Due to the prior-informed structured pruning, it is able to occupy the sweet spot of highly sparse at the same time biologically meaningful models.

Consistent with the literature, PathReg outperforms $L_0$ as well as C-NODE in terms of sparsity [1], we additionally find evidence that it also delivers more biologically meaningful results. Yet, IMP as well as prior-informed pruning approaches outperform PathReg by a large margin. The MSE of predicted gene expression of DASH is among the best, within one standard error of the best overall method. The only better approach is our suggested baseline, a combination of posthoc magnitude pruning of PHOENIX combined with additional finetuning (PINN+MP), which is, however, impractical as it requires to train and prune many PHOENIX models along a grid of sparsity levels (see App. B.9.3).

Visualizing the estimated against ground truth regulatory effects (i.e., functional relationships between variables), we observe that DASH captures the effects much better than competitors (see Fig. 3). Virtually all existing approaches discover spurious regulatory effects, whereas prior-informed pruning identify the main regulatory effects correctly. Moreover, with increasing levels of noise in the simulation, we observe that both BioPrune as well as PINN+MP start finding spurious dependencies, while DASH still recovers the overall structure well. While not perfect, as seemingly there are more dependencies than in the sparse ground truth, potentially introduced by correlations between features, DASH provides a sparse estimation of regulatory effects that most closely resembles the ground truth relationships among existing work.

**Pseudotime-ordered breast cancer samples** To investigate the performance of DASH on real data, we consider gene expression measurements from a cross-sectional breast cancer study [13]. This data of 198 breast cancer patients with measurements for 22000 genes has been preprocessed and ordered in pseudotime [57], which we use as basis for our experiments (cf App. B.7). Across methods, we observe that implicit sparsification methods generally perform poorly in terms of sparsity and accuracy of recovered relationships (see Tab. 2). While pruning-based sparsification approaches achieve greater sparsity and performance in predicted gene expression, with SparseFlow reaching the highest sparsity (95.7%) among all methods, the recovered biological relations are not better than random chance, which renders the underlying models useless for scientific discovery. DASH in contrast finds a comparably sparse network (92.7% sparsity) while having top of the line performance in terms of test MSE and high alignment with true biology (95.7% balanced accuracy). For this particular dataset, we observe that DASH primarily builds on the prior knowledge, not surprisingly performing similarly as BioPrune, which is our suggested baseline pruning approach taking only the prior into account. We will see for other real-world data that this weight of domain knowledge is highly task-specific and BioPrune yields sub-optimal results on different data.

To better understand whether the inferred gene regulatory dynamics align with meaningful biology, we additionally perform a pathway analysis (see App. B.8). Such pathway analysis are a standard approach for domain experts to distill information for example for therapeutic design. The genes that show the highest impact on the dynamics within the derived model are tested whether they enrich in a specific higher level biological pathways. For the top-20 most significantly enriched pathway per model (App. Fig. 5), we observe that in contrast to prior-informed methods, the existing pruning approaches show only very few significant pathways, consistent with our quantitative results on

Table 2: Results on breast cancer and yeast data. Balanced accuracy is based on reference gold standard experiments (transcription factor binding ChIP-seq) available for this data. DASH found optimal $\lambda$-values of $(0.995, 0.95)$ respectively $(0.75, 0.75)$ for breast cancer and yeast. * marks our suggested baselines and method, ✓ marks methods that use prior information for sparsification.

| Data | Breast cancer in pseudotime | | | Yeast cell cycle | | |
|---|---|---|---|---|---|---|
| Strategy | Sparsity | Bal. Acc. | MSE $(10^{-5})$ | Sparsity | Bal. Acc. | MSE $(10^{-2})$ |
| None/Baseline | 0.03% | 49.99% | 7.78 | 0.10% | 49.87% | **4.84** |
| $L_0$ | 10.77% | 50.15% | 7.90 | 34.43% | 48.43% | 5.33 |
| C-NODE | 11.20% | 50.01% | 8.06 | 10.89% | 50.04% | 4.87 |
| PathReg | 14.09% | 50.24% | 7.92 | 12.09% | 50.11% | 5.35 |
| PINN ✓ | 0.11% | 49.99% | 7.82 | 0.17% | 49.93% | 5.77 |
| DST | 67.02% | 50.42% | 7.78 | 77.80% | 49.92% | 5.18 |
| IMP | 36.02% | 50.34% | 7.77 | 83.22% | 49.99% | 5.46 |
| Iter. SynFlow | 91.93% | 49.37% | 7.78 | 85.65% | 49.57% | 5.41 |
| SparseFlow | **95.70**% | 49.70% | **7.76** | 95.22% | 49.89% | 5.38 |
| BioPrune *, ✓ | 93.44% | 95.67% | 7.80 | 94.69% | 79.23% | 5.94 |
| DASH *, ✓ | 92.71% | **95.69**% | **7.76** | **97.18**% | **88.43%** | 5.27 |
| PINN + MP *, ✓ | 92.00% | 54.02% | 7.79 | 95.01% | 55.39% | 6.09 |

inferred regulatory relations. Moreover, disease-relevant pathways such as TP53 activity or FOXO mediated cell death, both of which are highly relevant in cancer [43, 28], are only visible in models pruned with prior information. This provides evidence that pruning informed by a biological prior recovers biological signals that are relevant in the disease and which can not be picked up otherwise. Furthermore, we find Heme-signaling as a pathway uniquely identified as relevant in our approaches (cf. App. Fig. 5). Heme as a signaling molecule has key roles in the gene regulatory system [45], and turns out to have an anti-tumor role in breast cancer specifically [19]. Subsequent approaches pharmaceutically targeting Heme signaling showed success [30], with one of the key regulators affected being Bach1. To suggest further targets for e.g. combination treatment, we hence examined the top-5 regulatory factors in terms of weights in our estimated gene regulatory dynamics. These factors include PBX1 and FOXM1, for which a drug repurposing of existing compounds, such as [54], could lead to a potential new treatment for this specific cancer.

**Yest cell-cycle data**   We next consider real data of synchronized yeast cell [49] (see App. B.2 for training setup). We observe an overall trend similar to the breast cancer study in terms of achieved sparsity and balanced accuracy (cf. Tab. 2), with implicit sparsification methods generally finding significantly less sparse models and all methods that do not incorporate prior knowledge having inferred relationships that are not better than random chance. For this data, however, DASH finds an optimal lambda value that incorporates more data-specific knowledge ($\lambda = 0.75$) compared to the breast cancer study above. This shows the advantage of DASH over our BioPrune baseline model (prior-only pruning), as here we gain about 10% points in balanced accuracy over BioPrune for ChIP-seq validation data [23] and 2% points over BioPrune for an independent TF perturbation network curated to derive a "true" causal GRN [21] (see App. Tab. 8). We also retrieve a 3% points sparser model. Comparing inferred biological knowledge between BioPrune and DASH through a pathway analysis, we see that DASH recovers more significantly enriched pathways related to cell cycle processes (cf. App. Fig. 6).

**Cell differentiation in human bone marrow**   Lastly, we investigate the performance of DASH in an exploratory setting with single cell data of human bone marrow ordered in pseudotime [1]. Here, we are interested in better understanding the gene regulatory dynamics of blood cell differentiation, the process of hematopoietic stem cells specializing into cells taking over roles such as immune response (e.g., B- and T-cells). This process is called hematopoiesis. We follow the steps of [1] to first split samples (i.e., cells) into the three different lineages (paths of differentiation), and train separate models for each (see App. B.7). We will here focus on the analysis of the Erythroid lineage.

As before, DASH yields highly sparse (95%) networks, the most sparse among all competitors (App. Tab. 9). IMP shows similarly strong sparsification as DASH while PathReg achieves much less sparsity (14%). In terms of performance, all methods achieve similar MSE of predicted gene expression dynamics on the test set, meaning that even though much sparser, both DASH and IMP predict equally well as an order of magnitude more dense network. For this data, there are no gold-standard experiments for regulatory relations available, we hence focus on analysing the network topology. From the literature, we would expect sparser networks to be better align with biology [6]. DASH indeed shows the lowest out-degree in the inferred regulatory network, less than half of what IMP recovers. PathReg shows an order of magnitude larger average out-degree. To confirm the biological plausability, we again do a pathway analysis. DASH seems to find significant enrichment in biologically relevant pathways (App. Fig. 7) that can directly be linked to hematopoiesis, such as *heme signaling* or *RUNX1 regulates differentiation of hematopoietic stem cells*, which neither BioPrune nor SparseFlow—the only other method yielding a proper sparse model—could recover.

## 6 Discussion & Conclusion

We considered the problem of identifying sparse neural networks in the context of interpretability with a focus on the application to gene regulatory systems modeling. In domains such as biology and contexts when the true underlying systems are sparse, interpretability is key for experts, rendering the use of the common over-parametrized and complex neural network architectures difficult. Although NNs do not directly encode e.g. the regulatory relationship between genes its deep architecture is necessary to model complex functional relationships while ensuring stable learning. Yet, we can ensure that Recent advances in neural network pruning, such as those around the Lottery Ticket Hypothesis [18], promise sparse and well-performing models, yet, hardness results prove finding optimally sparse models to be challenging [42], which is also reflected by recent benchmarking results [16]. Our experiments confirmed that general pruning strategies provide sub-optimal sparsity, moreover, the underlying biological relationships are not properly reflected in the model. We proposed *to guide pruning by domain knowledge*, leveraging existing prior information to improve the interpretability and meaningfulness of pruned models.

In case studies on gene regulatory dynamic inference, a key task in molecular biology with high relevance in cancer research, we showed based on simulated as well as real world data that our method, DASH, in contrast to a wide range of state-of-the-art methods, is able to recover neural networks that are both very sparse and at the same time biologically meaningful, allowing for direct extraction of a sparse gene regulatory network. On real data, DASH not only better aligns with gold-standard experimental evidence of regulatory interactions, but also uniquely reflects the data-specific biological pathways, which can be used by domain experts to generate new insights.. It thus serves as a proof of concept that in critical domains, where interpretability is essential and domain knowledge exists, pruning can be heavily improved by alignment with prior knowledge. While our guided pruning approach is in principle agnostic to the type of neural network and task, we here focused on a specific case study that we deemed important. In the future, it would be interesting to apply DASH to different cases and domains, including other biomedical tasks, but also to physics or material sciences, where interpretability is also key and domain knowledge exists in the form of physical constraints and models. For any application, an important consideration to apply DASH is on the one hand the availability of prior knowledge, but on the other hand its quality; while we show here that even with incomplete and noisy prior knowledge we receive good results, factually wrong priors could steer the solution towards a wrong model. We hence assume that DASH will be of primary use in classical hard sciences mentioned above, with priors that stood the test of time over several decades. Another line of future work includes different architectural designs, such as convolution or attention mechanisms, where input-output relationships are less straightforward to project to across several layers.

In summary, we make a case for pruning informed by domain knowledge and provide evidence that such approaches can massively improve sparsity along with domain specific interpretability.

**Acknowledgements** RB received funding from the European Research Council (ERC) under the Horizon Europe Framework Programme (HORIZON) for proposal number 101116395 SPARSE-ML. JQ was supported by a grant from the US National Cancer Institute (R35CA220523) and additional funding from the National Human Genome Research Institute (R01HG011393).

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

# A    Supplementary results

## A.1    Synthetic data

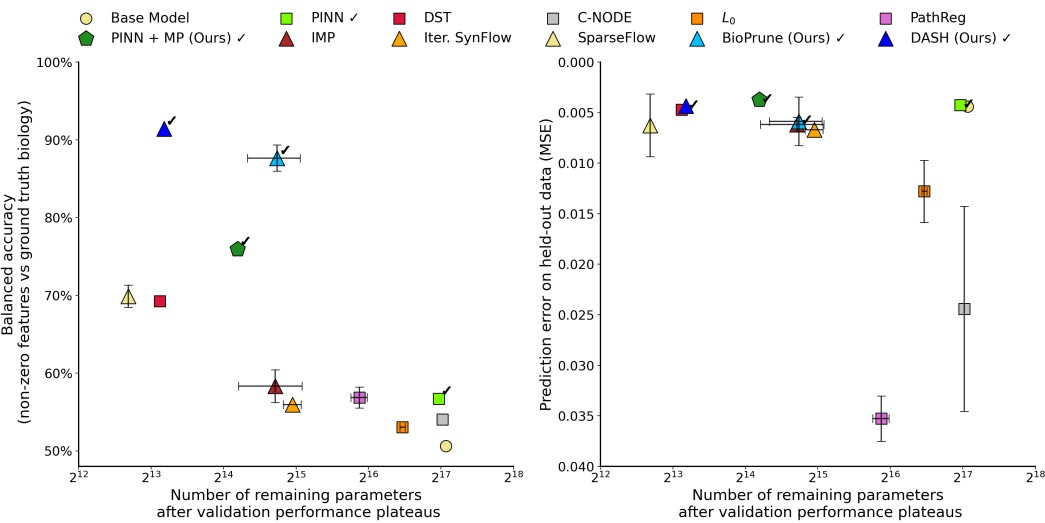

Figure 4: *SIM690 data with 5% noise.* We visualize performance of pruning strategies in comparison to original PHOENIX (baseline) in terms of achieved sparsity (x-axis) and balanced accuracy (y-axis) of the recovered gene regulatory network against the ground truth. Error bars are omitted when error is smaller than depicted symbol. Checkmarks (✓) are used to indicate methods that leverage prior information. Ideal models are in the top left quadrant; they recover the true, inherently sparse biological relationships.

Table 3: *Simulation study results – 0% noise.* We provide achieved model sparsity, balanced accuracy of inferred gene regulatory network, and MSE of predicted gene expression dynamics on test data at 0% noise level for SIM350 (the synthetic system of 350 genes). * marks our suggested baselines and method, ✓ marks methods that use prior information for sparsification.

|  | Strategy | Sparsity(%) | Bal. Acc.(%) | MSE ($10^{-3}$) | Epochs |
|---|---|---|---|---|---|
|  | None/Baseline [27] | $11.5 \pm 0.3$ | $54.8 \pm 0.6$ | $3.6 \pm 1.7$ | $69 \pm 2$ |
| Penalty-based (implicit) | $L_0$ [40] | $34.7 \pm 2.4$ | $61.3 \pm 0.1$ | $6.1 \pm 1.8$ | $119 \pm 43$ |
|  | C-NODE [2] | $10.7 \pm 0.2$ | $60.5 \pm 0.1$ | $\mathbf{1.9 \pm 0.5}$ | $213 \pm 5$ |
|  | PathReg [1] | $59.7 \pm 1.5$ | $64.2 \pm 0.8$ | $6.1 \pm 2.3$ | $213 \pm 7$ |
|  | PINN [27] ✓ | $11.3 \pm 0.3$ | $60.3 \pm 0.3$ | $2.3 \pm 0.4$ | $206 \pm 7$ |
|  | DST[38] | $94.3 \pm 0.5$ | $72.3 \pm 1.2$ | $4.2 \pm 1.4$ | $216 \pm 86$ |
| Pruning-based (explicit) | IMP [18] | $86.1 \pm 5.1$ | $63.2 \pm 1.7$ | $4.1 \pm 0.6$ | $251 \pm 7$ |
|  | Iter. SynFlow [59] | $79.1 \pm 2.1$ | $60.0 \pm 0.9$ | $5.8 \pm 1.6$ | $323 \pm 37$ |
|  | SparseFlow [37] | $\mathbf{95.8 \pm 0.3}$ | $72.8 \pm 0.7$ | $2.9 \pm 0.5$ | $195 \pm 16$ |
|  | BioPrune *, ✓ | $83.5 \pm 3.5$ | $88.0 \pm 0.5$ | $3.6 \pm 0.8$ | $94 \pm 22$ |
|  | DASH *, ✓ | $92.6 \pm 1.2$ | $\mathbf{91.1 \pm 1.2}$ | $\mathbf{1.9 \pm 0.6}$ | $164 \pm 22$ |
| Hybrid | PINN + MP *, ✓ | $90.0 \pm 0.01$ | $89.8 \pm 0.3$ | $2.6 \pm 0.5$ | $1788 \pm 77$ |

Table 4: *Simulation study results – 10% noise.* We provide achieved model sparsity, balanced accuracy of inferred gene regulatory network, and MSE of predicted gene expression dynamics on test data at 10% noise level for SIM350 (the synthetic system of 350 genes). * marks our suggested baselines and method, ✓ marks methods that use prior information for sparsification.

| | Strategy | Sparsity(%) | Bal. Acc.(%) | MSE ($10^{-3}$) | Epochs |
|---|---|---|---|---|---|
| | None/Baseline [27] | $1.3 \pm 0.3$ | $50.1 \pm 0.3$ | $\mathbf{3.5 \pm 0.03}$ | $55 \pm 5$ |
| Penalty-based (implicit) | $L_0$ [40] | $31.9 \pm 1.6$ | $50.2 \pm 0.3$ | $13.3 \pm 1.8$ | $156 \pm 11$ |
| | C-NODE [2] | $7.2 \pm 0.3$ | $50.5 \pm 0.3$ | $36.6 \pm 12.5$ | $189 \pm 11$ |
| | PathReg [1] | $47.8 \pm 1.8$ | $50.2 \pm 1.2$ | $12.7 \pm 1.4$ | $224 \pm 8$ |
| | PINN [27] ✓ | $2.9 \pm 0.5$ | $51.3 \pm 0.3$ | $5.5 \pm 1.9$ | $211 \pm 15$ |
| | DST[38] | $93.3 \pm 2.0$ | $67.2 \pm 2.6$ | $4.1 \pm 1.1$ | $286 \pm 33$ |
| Pruning-based (explicit) | IMP [18] | $79.8 \pm 0.1$ | $59.5 \pm 2.1$ | $6.7 \pm 1.5$ | $240 \pm 26$ |
| | Iter. SynFlow [59] | $79.7 \pm 1.3$ | $55.9 \pm 1.2$ | $7.8 \pm 0.03$ | $165 \pm 5$ |
| | SparseFlow [37] | $89.9 \pm 5.0$ | $63.8 \pm 3.0$ | $5.1 \pm 0.8$ | $142 \pm 13$ |
| | BioPrune *, ✓ | $79.7 \pm 1.3$ | $85.2 \pm 0.3$ | $5.6 \pm 0.3$ | $67 \pm 18$ |
| | DASH *, ✓ | $90.8 \pm 4.7$ | $\mathbf{85.4 \pm 4.2}$ | $5.8 \pm 1.3$ | $154 \pm 4$ |
| Hybrid | PINN + MP *, ✓ | $\mathbf{92.0 \pm 0.01}$ | $83.8 \pm 0.7$ | $4.1 \pm 1.8$ | $1813 \pm 94$ |

Table 5: *Comparison of systems with different number of genes $N_g$.* We provide achieved model sparsity, balanced accuracy of inferred gene regulatory network, and MSE of predicted gene expression dynamics on test data at 5% noise level for SIM350 and SIM690. * marks our suggested baselines and method, ✓ marks methods that use prior information for sparsification.

| $N_g$ | | Strategy | Sparsity(%) | Bal. Acc.(%) | MSE ($10^{-3}$) | Epochs |
|---|---|---|---|---|---|---|
| | | None/Baseline [27] | $7.5 \pm 0.1$ | $51.8 \pm 0.03$ | $3.0 \pm 0.4$ | $67 \pm 6$ |
| | Penalty-based (implicit) | $L_0$ [40] | $33.8 \pm 4.7$ | $55.0 \pm 0.5$ | $8.5 \pm 1.0$ | $134 \pm 37$ |
| | | C-NODE [2] | $6.2 \pm 0.5$ | $55.9 \pm 0.1$ | $2.8 \pm 0.6$ | $214 \pm 4$ |
| | | PathReg [1] | $56.5 \pm 1.5$ | $61.9 \pm 1.0$ | $8.0 \pm 1.8$ | $200 \pm 29$ |
| | | PINN [27] ✓ | $9.9 \pm 0.4$ | $58.6 \pm 0.7$ | $2.5 \pm 0.2$ | $160 \pm 7$ |
| | | DST[38] | $92.8 \pm 0.3$ | $71.9 \pm 0.5$ | $4.0 \pm 0.5$ | $244 \pm 61$ |
| 350 | Pruning-based (explicit) | IMP [18] | $81.9 \pm 6.6$ | $61.7 \pm 0.7$ | $4.7 \pm 1.1$ | $308 \pm 13$ |
| | | Iter. SynFlow [59] | $79.3 \pm 1.2$ | $58.4 \pm 0.6$ | $7.0 \pm 2.1$ | $271 \pm 38$ |
| | | SparseFlow [37] | $\mathbf{96.0 \pm 0.01}$ | $70.9 \pm 1.5$ | $3.6 \pm 0.6$ | $220 \pm 3$ |
| | | BioPrune *, ✓ | $83.5 \pm 1.9$ | $87.3 \pm 0.8$ | $2.6 \pm 0.9$ | $79 \pm 2$ |
| | | DASH *, ✓ | $94.6 \pm 1.2$ | $\mathbf{90.7 \pm 0.4}$ | $2.4 \pm 1.2$ | $192 \pm 24$ |
| | Hybrid | PINN + MP *, ✓ | $87.0 \pm 0.01$ | $82.4 \pm 0.2$ | $\mathbf{2.3 \pm 0.3}$ | $1721 \pm 50$ |
| | | None/Baseline [27] | $1.1 \pm 0.3$ | $50.6 \pm 0.1$ | $4.4 \pm 0.4$ | $188 \pm 17$ |
| | Penalty-based (implicit) | $L_0$ [40] | $34.9 \pm 1.1$ | $53.0 \pm 0.3$ | $12.8 \pm 3.1$ | $100 \pm 26$ |
| | | C-NODE [2] | $4.3 \pm 0.4$ | $54.0 \pm 0.2$ | $24.4 \pm 10.2$ | $210 \pm 2$ |
| | | PathReg [1] | $57.7 \pm 0.5$ | $57.4 \pm 0.2$ | $35.3 \pm 2.2$ | $244 \pm 17$ |
| | | PINN [27] ✓ | $7.7 \pm 0.1$ | $56.7 \pm 0.03$ | $4.3 \pm 0.3$ | $166 \pm 29$ |
| | | DST[38] | $94.1 \pm 0.2$ | $69.2 \pm 0.9$ | $4.7 \pm 0.4$ | $188 \pm 28$ |
| 690 | Pruning-based (explicit) | IMP [18] | $81.1 \pm 5.2$ | $58.3 \pm 2.1$ | $6.2 \pm 0.7$ | $319 \pm 24$ |
| | | Iter. SynFlow [59] | $77.6 \pm 1.4$ | $55.9 \pm 0.4$ | $6.7 \pm 0.05$ | $215 \pm 18$ |
| | | SparseFlow [37] | $\mathbf{95.8 \pm 0.3}$ | $69.9 \pm 1.5$ | $6.3 \pm 3.1$ | $205 \pm 10$ |
| | | BioPrune *, ✓ | $80.8 \pm 4.3$ | $87.7 \pm 1.7$ | $5.9 \pm 2.4$ | $71 \pm 23$ |
| | | DASH *, ✓ | $93.9 \pm 0.01$ | $\mathbf{91.4 \pm 0.2}$ | $4.3 \pm 0.5$ | $178 \pm 2$ |
| | Hybrid | PINN + MP *, ✓ | $87.0 \pm 0.01$ | $75.9 \pm 0.2$ | $\mathbf{3.7 \pm 0.2}$ | $1657 \pm 106$ |

Table 6: *Sensitivity of DASH to noise in prior.* To understand the impact of the quality of the prior knowledge on the performance of DASH, we show results for different levels of prior corruption in the synthetic data (SIM 350). We keep expression noise constant at 0% to understand the impact of prior corruption alone.

| Strategy | Prior corruption | Sparsity(%) | Bal. Acc.(%) | MSE ($10^{-3}$) |
|---|---|---|---|---|
| None/Baseline | *Does not use prior* | 11.5 | 54.8 | 3.6 |
| $L_0$ | *Does not use prior* | 34.7 | 61.3 | 6.1 |
| C-NODE | *Does not use prior* | 10.7 | 60.5 | 1.9 |
| PathReg | *Does not use prior* | 59.7 | 64.2 | 6.1 |
| DST | *Does not use prior* | 94.3 | 72.3 | 4.2 |
| IMP | *Does not use prior* | 86.1 | 63.2 | 4.1 |
| Iter. SynFlow | *Does not use prior* | 79.1 | 60.0 | 2.3 |
| SparseFlow | *Does not use prior* | 95.8 | 72.8 | 2.9 |
| | 0% | 11.3 | 60.3 | 2.3 |
| PINN | 20% | 12.4 | 60.8 | 3.1 |
| | 40% | 11.2 | 60.6 | 2.7 |
| | 0% | 83.5 | 88.0 | 3.6 |
| BioPrune | 20% | 80.9 | 81.5 | 7.6 |
| | 40% | 86.8 | 80.1 | 11.1 |
| | 0% | 92.6 | 91.1 | 1.9 |
| DASH | 20% | 92.4 | 86.2 | 6.7 |
| | 40% | 85.9 | 79.5 | 6.1 |

Table 7: *Prior-informed pruning on an MLP for simulated data.* We compare sparsification strategies on PHOENIX base model and a simple 2-layer MLP base model with ELU activations. We tested on SIM350 with 5% noise. Balanced Accuracy is included since ground truth regulatory structure is known.

| | PHOENIX base model | | | MLP base model (with ELU) | | |
|---|---|---|---|---|---|---|
| Strategy | Sparsity | Bal Acc | Test MSE ($10^{-3}$) | Sparsity | Bal Acc | Test MSE ($10^{-3}$) |
| None | 7.5% | 51.8% | 3.0 | 0% | 50.0% | 5.2 |
| $L_0$ | 33.8% | 55.0% | 8.5 | 29.6% | 51.2% | 8.1 |
| C-NODE | 6.2% | 55.9% | 2.8 | 48.8% | 50.3% | 5.3 |
| PathReg | 56.5% | 61.9% | 8.0 | 47.1% | 55.3% | 8.2 |
| DASH | 94.6% | 90.7% | 2.4 | 89.4% | 88.4% | 3.8 |

## A.2    Breast cancer data

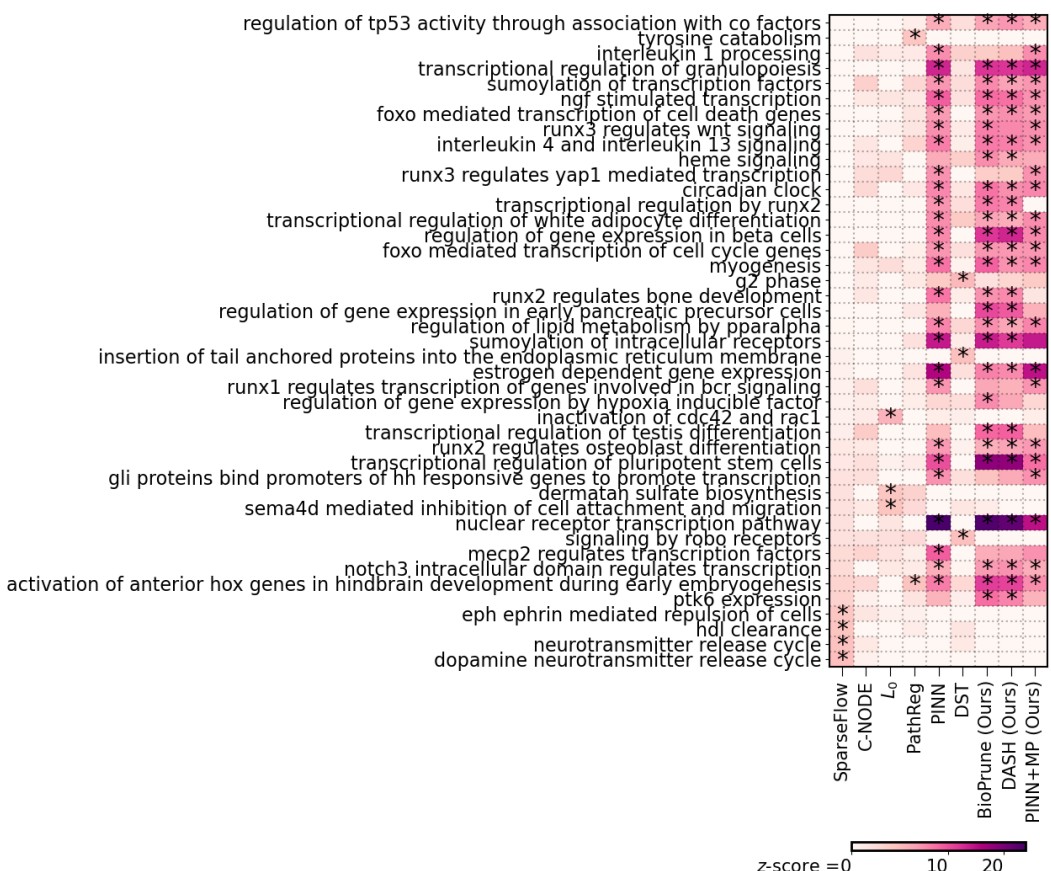

Figure 5: *BRCA pathway analysis.* We visualize the top-20 significant pathways for each method, showing the pathway z-score (x-axis) and indicate significant results after FWER correction (Bonferroni, p-value cutoff at .05) with *.

## A.3 Yeast data

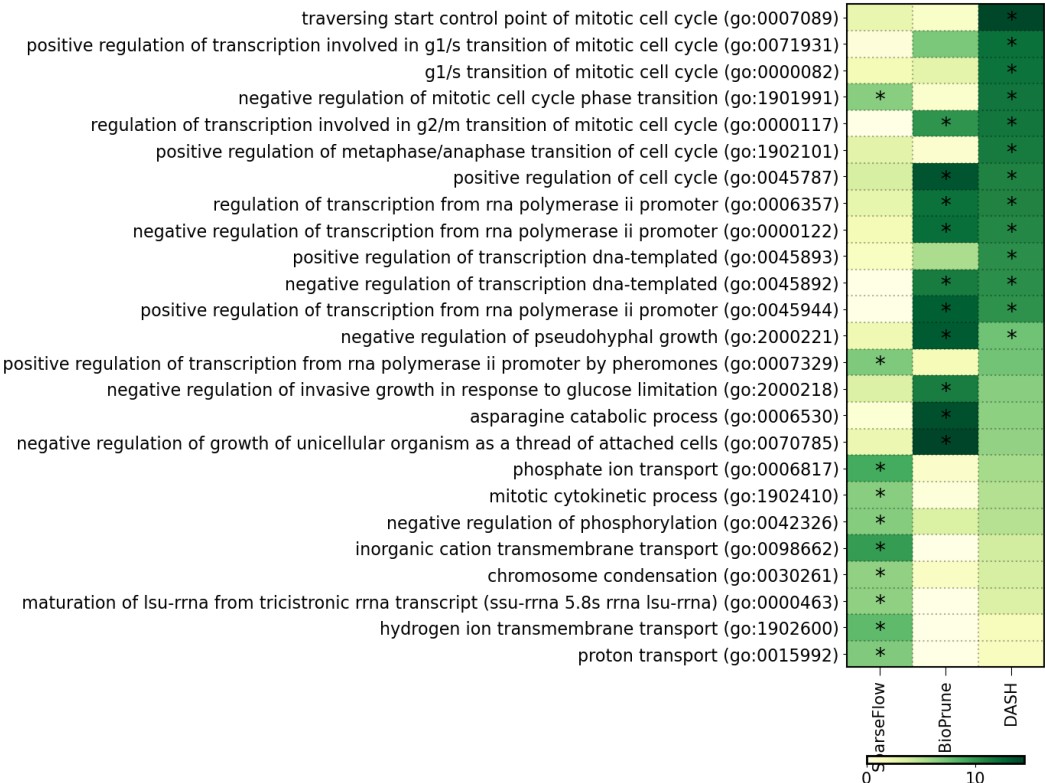

Figure 6: *Yeast pathway analysis.* We visualize the top-20 significant pathways for each method, showing the pathway z-score (x-axis) and indicate significant results after FWER correction (Bonferroni, p-value cutoff at .05) with *.

| Strategy | Sparsity | Bal. Acc. (ChipSeq) | Bal. Acc. (TF Perturb) | MSE ($10^{-2}$) |
|---|---|---|---|---|
| None/Baseline | 0.10% | 49.87% | 49.92% | **4.84** |
| $L_0$ | 34.43% | 48.43% | 49.28% | 5.33 |
| C-NODE | 10.89% | 50.04% | 50.17% | 4.87 |
| PathReg | 12.09% | 50.11% | 49.92% | 5.35 |
| PINN ✓ | 0.17% | 49.93% | 50.01% | 5.77 |
| DST | 77.80% | 49.92% | 50.33% | 5.18 |
| IMP | 83.22% | 49.99% | 48.45% | 5.46 |
| Iter. SynFlow | 85.65% | 49.57% | 49.77% | 5.41 |
| SparseFlow | 95.22% | 49.89% | 51.58% | 5.38 |
| BioPrune ∗, ✓ | 94.69% | 79.23% | 64.50% | 5.94 |
| DASH ∗, ✓ | **97.18%** | **88.43%** | **66.79%** | 5.27 |
| PINN + MP ∗, ✓ | 95.01% | 55.39% | 52.95% | 6.09 |

Table 8: *Balanced accuracies for experiments on yeast data.* Balanced accuracy is based on **1)** transcription factor binding ChIP-seq available for this data [23] and **2)** A TF perturbation network created by Hackett *et al.* based on their TF perturbation experiments [21]. * marks our suggested baselines and method, ✓ marks methods that use prior information for sparsification.

## A.4 Bone marrow data

Table 9: *Results on Hematopoesis data for the Erythroid lineage.* We give sparsity of pruned model and test MSE on predicted gene expression dynamics. No reference gene regulatory network is available to compute the accuracy of the recovered network we hence resort to reporting the average out-degree of nodes in the recovered network. DASH found an optimal $\lambda$-value of $(0.80, 0.80)$. * marks our suggested baselines and method, ✓ marks methods that use prior information for sparsification.

| Strategy | Sparsity | OutDeg | Test MSE ($10^{-4}$) |
|---|---|---|---|
| None/Baseline | 0.25% | 529 | **2.12** |
| $L_0$ | 12.03% | 518 | 2.14 |
| C-NODE | 1.33% | 529 | 2.17 |
| PathReg | 13.94% | 522 | **2.12** |
| PINN ✓ | 5.93% | 529 | 2.14 |
| DST | 90.54% | 328 | 2.23 |
| IMP | 73.00% | 404 | 2.26 |
| Iter. SynFlow | 82.28% | 350 | 2.18 |
| SparseFlow | 94.44% | 123 | **2.12** |
| BioPrune $*$, ✓ | 87.09% | 319 | 2.14 |
| DASH $*$, ✓ | 95.95% | 54 | **2.12** |
| PINN + MP $*$, ✓ | 92.00% | 218 | 2.14 |

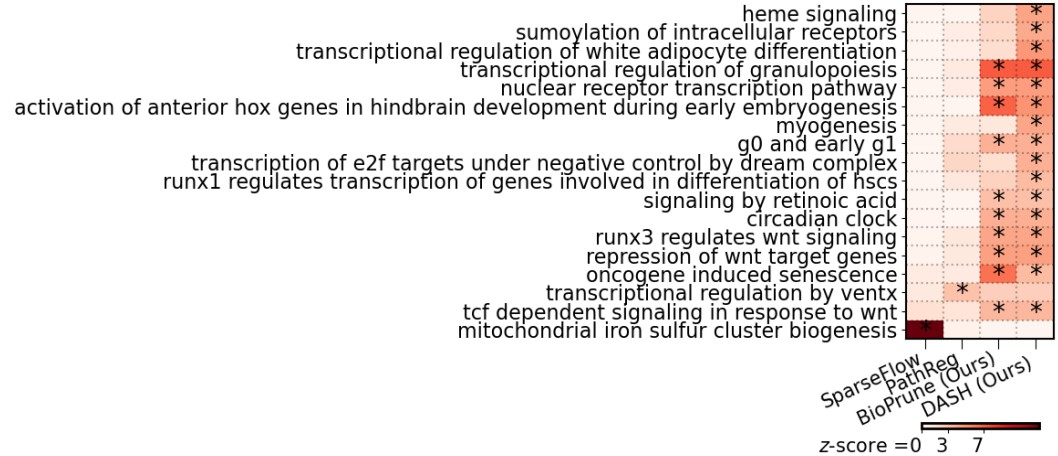

Figure 7: *Hematopoesis pathway analysis.* We visualize the top-20 significantly enriched pathways on the Erythroid lineage. For each method (x-axis) we show the pathway z-score and indicate significant results after FWER correction (Bonferroni, p-value cutoff at .05) with *.

Table 10: *Prior-informed pruning on an MLP for bone marrow data*. We compare sparsification strategies on PHOENIX base model and a simple 2-layer MLP base model with ELU activations. We tested on Erythroid lineage of the bone marrow data.

| | PHOENIX base model | | MLP base model (with ELU) | |
|---|---|---|---|---|
| Strategy | Sparsity | Test MSE ($10^{-4}$) | Sparsity | Test MSE ($10^{-4}$) |
| None | 0.25% | 2.12 | 0.00% | 2.11 |
| $L_0$ | 12.03% | 2.14 | 8.41% | 2.29 |
| C-NODE | 1.33% | 2.17 | 2.55% | 2.22 |
| PathReg | 13.94% | 2.12 | 21.48% | 2.15 |
| DASH | 95.95% | 2.12 | 84.06% | 2.13 |

# B    Supplementary methods

## B.1    Synthetic data generation

The purpose of simulation based data is so that the the underlying dynamical system that produced the this gene expression was known. Do this end, we closely follow the steps outlined by the simulation pipeline provided by [27] to generate reliable synthetic time-series gene expression data from two ground truth gene regulatory networks $G_{350}$ and $G_{690}$ consisting of 350 and 690 genes, respectively.

The pipeline adapts `SimulatorGRN` [4] to generate from two synthetic *S. cerevisiae* gene regulatory systems (SIM350 and SIM690, consisting of 350 and 690 genes respectively). For every noise setting $\in \{0\%, 5\%, 10\%\}$, the connectivity structure of each *in silico* system is used to synthesize 160 noisy expression trajectories for each gene across $t \in T = \{0, 2, 3, 7, 9\}$. We split up the trajectories into training (88%), validation (6% for tuning $\lambda$), and testing (6%). Since the average simulated expression value is $\approx 0.5$, adding Gaussian noise of $\mathcal{N}(0, \sigma^2)$ using $\sigma \in \{0, \frac{1}{40}, \frac{1}{20}\}$ corresponds roughly to average noise levels of $\{0\%, 5\%, 10\%\}$.

## B.2    Setup for model training

### B.2.1    Model complexity

Since the number of genes $k$ in each problem is different, the number of neurons $m$ in PHOENIX's hidden layer is chosen to roughly scale with this $k$ according to the original paper [27]:

- SIM350: $k = 350$, $m = 40$
- SIM690: $k = 690$, $m = 50$
- Bone marrow data: $k = 529$, $m = 50$
- Yeast data: $k = 3551$, $m = 120$
- Breast cancer data: $k = 11165$, $m = 300$

### B.2.2    Initialization and optimizers

For initialization values for each of $NN_{sums}$, $NN_{prods}$, $NN_{\Sigma combine}$, and $NN_{\Pi combine}$, as well as that of $v_i$s we choose the default provided by the PHOENIX implementation [27]. The ODESolver (dopri5) and optimizer (Adam) are also chosen as the PHOENIX defaults across all experiments.

### B.2.3    Pruning details

We use iterative **pruning schedules** that are initially very aggressive and then become much more gradual. We found this approach to achieve high sparsity without adversely affecting the training dynamics (and subsequently the validation performance).

- SIM350: prune 70% at epoch 3, and then 10% every 10 epochs
- SIM690: prune 70% at epoch 3, and then 10% every 10 epochs
- Bone marrow data: prune 70% at epoch 3, and then 10% every 10 epochs
- Yeast data: prune 90% at epoch 10, and then 10% every 20 epochs
- Breast cancer data: prune 90% at epoch 10, and then 10% every 20 epochs

**Weight normalization for DASH pruning scores**    As described in Section 3, the weight matrices of PHOENIX need to be normalized to $|\widetilde{W_\Sigma^{(t)}}|$, $|\widetilde{W_\Pi^{(t)}}|$, $|\widetilde{U_\Sigma^{(t)}}|$, and $|\widetilde{U_\Pi^{(t)}}|$ in the formula for calculating DASH pruning scores $\Omega_\Sigma$, $\Omega_\Pi$, $\Psi_\Sigma$, $\Psi_\Pi$. We perform the following normalizations:

- For $|\widetilde{W_\Sigma^{(t)}}|$ we simply normalize by taking elementwise absolute values of $W_\Sigma^{(t)}$ and dividing all entries by the overall sum of absolute values.

- For $|\widetilde{\boldsymbol{W}_{\boldsymbol{\Pi}}^{(t)}}|$ we approach similarly, with the only modification that the weights are element-wise exponentiated instead of elementwise absolute value, given that $\widetilde{\boldsymbol{W}_{\boldsymbol{\Pi}}^{(t)}}$ operates on the log-space.

- For $|\widetilde{\boldsymbol{U}_{\boldsymbol{\Sigma}}^{(t)}}|$ and $|\widetilde{\boldsymbol{U}_{\boldsymbol{\Pi}}^{(t)}}|$ we approach again similarly, with the important modification that the gene-specific multipliers (from Section B.10.2) are row-wise multiplied into the weight matrices prior to normalization. This allows the effect of gene multipliers to appropriately be considered when performing pruning.

### B.2.4  Learning rates

The learning rate is used as the PHOENIX default of $10^{-3}$. We reduce the learning rate by 10% every 3 epochs, unless the validation set performance shows reasonable improvement. **Importantly**, we reset the learning rate back to $10^{-3}$ immediately after a pruning step is completed, thereby allowing the newly sparsified model to start learning with a higher learning rate.

### B.2.5  Stopping criteria

We train for up to 500 epochs on an AWS c5.9xlarge instance, where each epoch consisted of the entire training set being fed to the model, preceded by any pruning step that is prescribed by the pruning schedule. Training is terminated if the validation set performance fails to improve in 40 consecutive epochs. Upon training termination, we have obtained a model that has been iteratively sparsified to an extent that fails to improve the validation set performance. Hence this training procedure **automatically finds an optimal sparsity level** using the validation set.

### B.3  Prior knowledge to obtain DASH pruning scores

As mentioned in Section 3, DASH can leverage prior matrices $\boldsymbol{P}$ and $\boldsymbol{C}$ to inform its pruning score. We use the following in our experiments:

- SIM350 and SIM690:
    - for synthetic experiments we choose $\boldsymbol{P} = \boldsymbol{A}^{\sigma\%}$ to be noisy/corrupted versions (see B.3.1) of the adjacency matrices of ground truth networks $G_{350}$ and $G_{690}$ to reflect that transcription factor binding to target genes can itself be a noisy process in real life. A 1 $\boldsymbol{A}^{\sigma\%}$ represents prior knowledge of an interaction existing between two genes, and a 0 represented no interaction.
    - For $\boldsymbol{C}$ we use the outer product $\boldsymbol{C} = \boldsymbol{A}^{\sigma\%}(\boldsymbol{A}^{\sigma\%})^{\mathsf{T}}$, to represent prior knowledge of **co**regulation. We again applied the corruption/missepecification procedure from B.3.1 so that $\boldsymbol{C}$ is also noisy.

- Breast cancer data:
    - For the prior domain knowledge, we set $\boldsymbol{P} = \boldsymbol{W_0}$, where $\boldsymbol{W_0}$ was a motif map derived from the human reference genome, for the breast tissue specifically, which we obtained through GRAND [3]. $W_0$ is a binary matrix with $\boldsymbol{W_0}_{i,j} \in \{0,1\}$ where 1 indicates a likely occurence of a TF sequence motif in the promoter of the target gene, and hence indicating a putative interaction. More simply put, it indicates that whether there is (likely) a binding interface for the protein close to the target gene.
    - Based on information from the STRING database [58], we obtained a protein-protein interaction matrix (PPI) which could use to operationalize our $\boldsymbol{C}$ matrix, since a PPI is a again a binary matrix that is suggestive of which transcription factors have combined (or coregulatory) effects. In a nutshell, the STRING database is a graph with proteins as vertices and knowledge about interactions between two proteins in the graph specifying edges. We set an entry $P_i j$ to 1 if the experimental evidence score on the edge between protein $i$ and $j$ is larger than $0.6$, and set $P_i j$ to 0 otherwise.

- Bone marrow data:
    - For the prior domain knowledge, we followed a similar strategy as the breast cancer analysis, and set $\boldsymbol{P}$ and $\boldsymbol{C}$ based on the motif map and PPI matrix used in [61]. We

appropriately subsetted $P$ and $C$ to only be limited to the $k = 529$ genes that were selected by the PathReg authors [1] in the analysis.

- Yeast data:
    - For prior domain knowledge model we set $P$ to reflect the regulatory network structure of a motif map. The map is based on predicted binding sites for 204 yeast transcription factors (TFs) [23]. These data include 4360 genes with tandem promoters. 3551 of these genes are also covered on the yeast cell cycle gene expression array. 105 total TFs in this data set target the promoter of one of these 3551 genes. The motif map between these 105 TFs and 3551 target genes provides the adjacency matrix $A$ of 0s and 1s, representing whether or not a prior interaction is likely between TF and gene.
    - We set $C$ to be the PPI matrix used for the same data in the PANDA paper [20]. We appropriately subsetted $C$ to only be limited to the $k = 3551$ genes that were in the data.

### B.3.1 Creating corrupted/misspecified prior models for synthetic data

For each noise level $\sigma_\% \in \{0\%, 5\%, 10\%\}$ in our *in silico* experiments, we created a shuffled version of $G_{350}$ (and similarly $G_{690}$) where we shuffled $\sigma_\%$ of the edges by relocating those edges to new randomly chosen origin and destination genes within the network. This yielded the shuffled network $G_{350}^{\sigma_\%}$ (and similarly $G_{690}^{\sigma_\%}$) with corresponding adjacency matrix $A^{\sigma_\%}$. We additionally performed sensitivity analyses using $\sigma_\% \in \{20\%, 40\%\}$ to investigate the effect of **even** higher levels of prior corruption. We then used $A^{\sigma_\%}$ to obtain "corrupted" $P$ and $C$ as described in B.3.

## B.4 Validation and testing

The choice of $\boldsymbol{\lambda} = (\lambda_1, \lambda_2)$ is important for optimally combining prior information with model weights. Hence we implement a $K$-fold cross validation approach to choose $\boldsymbol{\lambda}$. Test set performance is measured as the mean squared error between predictions and held-out expression values in the test set.

## B.5 Measuring biological alignment of sparsified models

To validate biological alignment of trained and sparsified models, we extracted GRNs from each models (as explained in B.10.4), and compared back to the **validation networks**. Specifically, once we extracted a GRN from the trained model, we looked at how well 0s vs non-zeros in that network aligned with 0s vs non-zeros in the validation network. Our comparison metric was **balanced accuracy**, which is the average of the true positive and true negative rates. The validation networks were as follows:

- SIM350, SIM690: We used the ground truth networks $G_{350}$ and $G_{690}$.
- Breast cancer data: We used ChIP-seq data from the MCF7 cell line (breast cancer) in the ReMap2018 database [10] to create a validation network of TF-target interactions.
- Bone marrow data: As also noted by [1], validation network was not available, so we resorted to the pathway analysis.
- Yeast data: We used two kinds of validation networks
    1. ChIP-seq data [23] to create a network of TF-target interactions, and used this as a validation network to test explainability. The targets of transcription factors in this ChIP-chip data set were filtered using the criterion $p < 0.001$.
    2. A TF perturbation network created by Hackett *et al.*, who fit dynamical systems to their TF perturbation experiments [21].

## B.6 Strategy to potentially extend DASH to arbitrary number of layers

Supposing we only have access to prior knowledge in the form of putative prior effect sizes between the $n$ inputs and $o$ outputs $P \in \mathbb{R}^{o \times n}$. Then, for an NN with $L - 1$ layers $W_1 \in \mathbb{R}^{m_1 \times n}, W_2 \in \mathbb{R}^{m_2 \times m_1}, \ldots, W_L \in \mathbb{R}^{o \times m_L}$, we can adopt a strategy where we consider the pruning scores to be fixed for all but one layer.

Since the product $\boldsymbol{W}_L^{(t)} \ldots \boldsymbol{W}_2^{(t)} \cdot \boldsymbol{W}_1^{(t)} \in \mathbb{R}^{o \times n}$ represents the overall flow of information from inputs to outputs at epoch $t$, we surmise that $\boldsymbol{\Omega}_L^{(t)} \ldots \boldsymbol{\Omega}_2^{(t)} \cdot \boldsymbol{\Omega}_1^{(t)} \in \mathbb{R}^{o \times n}$ should reflect $\boldsymbol{P}$. We can thus prune as follows:

1. Starting with the last layer, we **fix** the pruning scores of all other layers and compute as follows:

$$\boldsymbol{\Omega}_L^{(t)} := (1 - \lambda_L) | \widetilde{\boldsymbol{W}_L^{(t)}} | + \lambda_L \left| \boldsymbol{P} \cdot \text{PInv}_R \left( \boldsymbol{\Omega}_{L-1}^{(t-1)} \ldots \boldsymbol{\Omega}_2^{(t-1)} \boldsymbol{\Omega}_1^{(t-1)} \right) \right|.$$

2. For the middle layers $l \in \{L-1, L-2, \ldots, 3, 2\}$, we do:

$$\boldsymbol{\Omega}_l^{(t)} := (1 - \lambda_l) | \widetilde{\boldsymbol{W}_l^{(t)}} | + \lambda_l \left| \text{PInv}_L \left( \boldsymbol{\Omega}_L^{(t)} \ldots \boldsymbol{\Omega}_{l+2}^{(t)} \boldsymbol{\Omega}_{l+1}^{(t)} \right) \cdot \boldsymbol{P} \cdot \text{PInv}_R \left( \boldsymbol{\Omega}_{l-1}^{(t-1)} \ldots \boldsymbol{\Omega}_2^{(t-1)} \boldsymbol{\Omega}_1^{(t-1)} \right) \right|.$$

3. The first layer can be pruned using:

$$\boldsymbol{\Omega}_1^{(t)} := (1 - \lambda_1) | \widetilde{\boldsymbol{W}_1^{(t)}} | + \lambda_1 \left| \text{PInv}_L \left( \boldsymbol{\Omega}_L^{(t)} \ldots \boldsymbol{\Omega}_3^{(t)} \boldsymbol{\Omega}_2^{(t)} \right) \cdot \boldsymbol{P} \right|.$$

### B.7 Processing steps for real data

#### B.7.1 Breast cancer

The original data set comes from a cross-sectional breast cancer study (GEO accession **GSE7390** [13]) consisting of microarray expression values for 22000 genes from 198 breast cancer patients, that is sorted along a pseudotime axis. We note that the same data set was also ordered in pseudotime by [57] in the PROB paper. For consistency in pseudotime inference, we obtained the same version of this data that was already preprocessed and sorted by PROB. We normalized the expression values to be between 0 and 1. We limited our analysis to the genes that had measurable expression and appeared in the aforementioned motif map and PPI matrices. This resulted in a pseudotrajectory of expression values for 11165 genes across 186 patients. We removed a contiguous interval of expression across 8 time points for testing (5%), and split up the remaining 178 time points into training (170, 90%) and validation for tuning $\lambda_{\text{prior}}$ (8, 5%).

#### B.7.2 Yeast

GPR files were downloaded from the Gene Expression Omnibus (accession **GSE4987** [49]), and consisted of two dye-swap technical replicates measured every five minutes for 120 minutes. Each of two replicates were separately ma-normalized using the `maNorm()` function in the `marray` library in `R/Bioconductor` [63]. The data were batch-corrected [29] using the `ComBat()` function in the `sva` library [33] and probe-sets mapping to the same gene were averaged, resulting in expression values for 5088 genes across fifty conditions. Two samples (corresponding to the 105 minute time point) were excluded for data-quality reasons, as noted in the original publication, and genes without motif information were then removed, resulting in an expression data set containing 48 samples (24 time points in each replicate) and 3551 genes.

#### B.7.3 Bone marrow

The data is originally from [41] (GEO accession code = **GSE194122**). The cleaning, preprocessing, and pseudotime analysis and was appropriately performed by [1] in the PathReg paper, and made publicly available, allowing us to access the processed version. Importantly, [1] split up the data into 3 different lineages (Erythroid, Monocyte, and B-Cell), and we fit a separate PHOENIX model on each lineage. The set contains 5 separate batches of data for each lineage, we used 1 for training (batch S1D2), 1 for validation (batch S1D1) and 3 for testing (batches S1D1, S2D4, and S3D6).

### B.8 Pathway analyses for breast cancer, yeast, and bone marrow datasets

We followed very closely the steps below from the Methods section of the PHOENIX paper [27] in order to compute pathway scores, with the only difference that we compute scores between different sparsification strategies `Pru`.

### B.8.1 Gene influence scores

Given $\mathcal{M}_{\texttt{Pru}}$ a PHOENIX model trained on a dataset consisting of $k$ genes, and sparsified using the pruning strategy $\texttt{Pru}$ (for $\texttt{Pru} \in \{\texttt{DASH}, \texttt{IMP}, \texttt{C-NODE}, \dots, \texttt{PathReg}\}$), we performed perturbation analyses to compute gene influence scores $\mathcal{IS}_{\texttt{Pru},j}$. We randomly generated 200 initial ($t = 0$) expression vectors via i.i.d standard uniform sampling $\{\boldsymbol{g}(\boldsymbol{0})_r \in \mathbb{R}^k\}_{r=1}^{200}$. Next, for each gene $j$ in $\mathcal{M}_{\texttt{Pru}}$, we created a perturbed version of these initial value vectors $\{\boldsymbol{g^j}(\boldsymbol{0})_r\}_{r=1}^{200}$, where only gene $j$ was perturbed in each unperturbed vector of $\{\boldsymbol{g}(\boldsymbol{0})_r\}_{r=1}^{200}$. We then fed both sets of initial values into $\mathcal{M}_{\texttt{Pru}}$ to obtain two sets of predicted trajectories $\{\{\widehat{\boldsymbol{g}}(\boldsymbol{t})_r \in \mathbb{R}^k\}_{t \in T}\}_{r=1}^{200}$ and $\{\{\widehat{\boldsymbol{g}^j}(\boldsymbol{t})_r \in \mathbb{R}^k\}_{t \in T}\}_{r=1}^{200}$ across a set of time points $T$. We calculated influence as the average absolute difference between the two sets of predictions, that represented how changes in initial ($t = 0$) expression of gene $j$ affected subsequent ($t > 0$) predicted expression of all other genes in the $\texttt{Pru}$-dimensional system

$$\mathcal{IS}_{\texttt{Pru},j} = \frac{1}{200} \sum_{r=1}^{200} \left[ \frac{1}{|T|} \sum_{\substack{t \in T \\ t \neq 0}} \left( \frac{1}{k} \sum_{\substack{i=1 \\ i \neq j}}^{k} |\widehat{g}_i(t)_r - \widehat{g}_i^j(t)_r| \right) \right].$$

### B.8.2 Pathway influence scores

Having computed gene influence scores $\mathcal{IS}_{\texttt{Pru},j}$ for each gene $j$ in each dynamical system of dimension $k$ genes sparsified with method $\texttt{Pru}$, we translated these gene influence scores into pathway influence scores. We used the Reactome pathway data set, GO biological process terms, and GO molecular function terms from MSigDB [36], that map each biological pathway/process, to the genes that are involved in it. For each system sparsified by $\texttt{Pru}$, we obtained the pathway ($p$) influence scores ($\mathcal{PS}_{\texttt{Pru},p}$) as the sum of the influence scores of all genes involved in pathway $p$

$$\mathcal{PS}_{\texttt{Pru},p} = \sum_{j \in p} \mathcal{IS}_{\texttt{Pru},j}.$$

We statistically tested whether each pathway influence score is higher than expected by chance using empirical null distributions. We randomly permuted the gene influence scores across the genes to recompute "null" values $\mathcal{PS}^0_{\texttt{Pru},p}$. For each pathway, we performed $Q = 1000$ permutations to obtain a null distribution $\{\mathcal{PS}^0_{\texttt{Pru},p,q}\}_{q=1}^{Q}$ that can be compared to $\mathcal{PS}_{\texttt{Pru},p}$. We could then compute an empirical $p$-value as $p = \frac{1}{Q} \sum_{q=1}^{Q} \mathbb{I}_{\mathcal{PS}^0_{\texttt{Pru},p,q} > \mathcal{PS}_{\texttt{Pru},p}}$, where $\mathbb{I}$ is the indicator function. Finally, we used the mean ($\mu_{0(\texttt{Pru},p)}$) and variance ($\sigma^2_{0(\texttt{Pru},p)}$) of the null distribution $\{\mathcal{PS}^0_{\texttt{Pru},p,q}\}_{q=1}^{Q}$ to obtain and visualize pathway $z$-scores that are now comparable across pathways ($p$) and sparsification strategies ($\texttt{Pru}$)

$$z_{(\texttt{Pru},p)} = \frac{\mathcal{PS}_{\texttt{Pru},p} - \mu_{0(\texttt{Pru},p)}}{\sqrt{\sigma^2_{0(\texttt{Pru},p)}}}.$$

## B.9 Implementation details for other sparsification strategies on the PHOENIX architecture

### B.9.1 Iterative magnitude pruning

As discussed in Section 3, IMP can be operationalized as a special case of DASH by setting $\lambda_1 = \lambda_2 = 0$.

### B.9.2 PINN

This is simply the PHOENIX model equipped with the prior-informed loss term. This loss-term in the original PHOENIX paper [27] is inspired by Physics-informed neural networks (PINNs).

### B.9.3 PINN + MP

Once a PHOENIX model (**trained including the prior-informed loss term, i.e. the PINN term**) is fully trained (without any pruning), we inspect the trained model and pruned the lowest $p\%$ of parameters in each of $\boldsymbol{W_\Sigma}, \boldsymbol{W_\Pi}, \boldsymbol{U_\Sigma}, \boldsymbol{U_\Pi}$ based on on the **normalized** weights (see B.2) to 0. We then fine-tune (i.e retrain **without** training the pruned parameters) this $p\%$ sparsified model and

calculate its performance on the validation set. We repeat this process for a grid of values for $p \in \{0.50, 0.75, 0.83, 0.87, 0.90, 0.92, 0.95, 0.97, 0.99\}$. The validation set can then inform the best value of $p$. We repeated this entire procedure 3 times, so that we could apply the 1 standard error rule [24] and choose the optimal $p$ as the **sparsest** fine-tuned model whose validation MSE is within 1 standard error of lowest obtained average validation MSE.

### B.9.4 Penalty based methods

C-NODE, PathReg, and $L_0$ implementations were obtained from the code associated with the PathReg paper [1]. We adapted the code so that the base NN architecture was exactly that of the PHOENIX model, including an implementation of the gene-specific multipliers (from Section B.10.2). Finally, we tuned the relevant parameters $\lambda_0$ and $\lambda_1$ in the objective function using the validation set.

The code for DST was obtained from: https://github.com/junjieliu2910/DynamicSparseTraining

### B.10 A brief overview of PHOENIX NeuralODE model

The following are adapted from [27] and provided here for the reader's convenience.

#### B.10.1 Neural ordinary differential equations

NeuralODEs [8] learn dynamical systems by parameterizing the underlying derivatives with neural networks: $\frac{dg(t)}{dt} = f(g(t), t) \approx \text{NN}_\Theta(g(t), t)$. Given an initial condition $g(t_0)$, the output $g(t_i)$ at any given time-point $t_i$ can now be approximated using a numerical ODE solver of adaptive step size:

$$\widehat{g(t_1)} = g(t_0) + \int_{t_0}^{t_1} \text{NN}_\Theta(g(t), t) \, dt.$$

A loss function $L\Big(g(t_1) ; g(t_0) + \int_{t_0}^{t_1} \text{NN}_\Theta(g(t), t) \, dt\Big)$ is then optimized for $\Theta$ via back propagation, using the adjoint sensitivity method [7] to carry the backpropagation through the integration steps of the ODESolver.

#### B.10.2 PHOENIX - overview

PHOENIX models gene expression dynamics using NeuralODEs. Notably, for an expression vector of $r$ genes, PHOENIX models both additive and multiplicative regulatory effects using two parallel linear layers with $m$ neurons each: $\text{NN}_{sums}$ (with weights $\boldsymbol{W}_\Sigma \in \mathbb{R}^{m \times r}$, and biases $\boldsymbol{b}_\Sigma \in \mathbb{R}^m$) and $\text{NN}_{prods}$ ($\boldsymbol{W}_\Pi \in \mathbb{R}^{m \times r}$, $\boldsymbol{b}_\Pi \in \mathbb{R}^m$). Here, $\text{NN}_{sums}$ and $\text{NN}_{prods}$ are equipped with activation functions that model the Hill equation

$$\phi_\Sigma(x) = \frac{x - 0.5}{1 + |\, x - 0.5\,|} \quad \text{and} \quad \phi_\Pi(x) = \log\Big(\phi_\Sigma(x) + 1\Big).$$

The Hill equation is a classical formula in biochemistry that models molecular binding in dependence of concentration. This results in outputs of the two parallel layers

$$c_\Sigma(g(t)) = \boldsymbol{W}_\Sigma \phi_\Sigma(g(t)) + \boldsymbol{b}_\Sigma \quad \text{and}$$
$$c_\Pi(g(t)) = \exp \circ (\boldsymbol{W}_\Pi \phi_\Pi(g(t)) + \boldsymbol{b}_\Pi).$$

As shown above, $\phi_\Pi(x)$ yields the output of $\text{NN}_{prods}$ in the log space and is subsequently exponentiated in $c_\Pi$ to represent multiplicative effects in the linear space. The outputs $c_\Sigma(g(t))$ and $c_\Pi(g(t))$ are then separately fed into two more parallel linear layers $\text{NN}_{\Sigma combine}$ (with weights $\boldsymbol{U}_\Sigma \in \mathbb{R}^{r \times m}$) and $\text{NN}_{\Pi combine}$ ($\boldsymbol{U}_\Pi \in \mathbb{R}^{r \times m}$), respectively. The outputs of $\text{NN}_{\Sigma combine}$ and $\text{NN}_{\Pi combine}$ are summed to obtain

$$c_\cup(g(t)) = \boldsymbol{U}_\Sigma c_\Sigma(g(t)) + \boldsymbol{U}_\Pi c_\Pi(g(t)).$$

Finally, PHOENIX includes gene-specific multipliers $\boldsymbol{v} \in \mathbb{R}^r$ for modeling steady states of genes that do not exhibit any temporal variation $\big(\frac{dg_i(t)}{dt} = 0, \forall t\big)$. Accordingly, the output for each gene $i$ in $c_\cup(g(t))$ is multiplied with $\text{ReLU}(v_i)$ in the final estimate for the local derivative

$$\text{NN}_\Theta(g(t), t) = \text{ReLU}(\boldsymbol{v}) \odot \Big[c_\cup(g(t)) - g(t)\Big].$$

Although PHOENIX achieves some sparsity in its weight matrices ($\boldsymbol{W}_\Sigma, \boldsymbol{W}_\Pi, \boldsymbol{U}_\Sigma, \boldsymbol{U}_\Pi$) **without any external sparsification strategy**, the achieved sparsity level is, however, at most 12% (see Figure 2, and Tables 1, 3, 4).

### B.10.3  Prior knowledge incorporation in base PHOENIX model itself

PHOENIX has the option to promote the NeuralODE to flexibly align with structural domain knowledge, while still explaining the observed gene expression data. This is operationalized via a modified loss function

$$\mathcal{L}_{\text{mod}}\Big(\boldsymbol{g}(t_1), \widehat{\boldsymbol{g}(t_1)}\Big) = \tau \overbrace{L\Big(\boldsymbol{g}(t_1)\,;\boldsymbol{g}(t_0) + \int_{t_0}^{t_1} \text{NN}_\Theta(\boldsymbol{g}(t_1), t)\,dt\Big)}^{\text{loss based on matching observed gene expression data}}$$

$$+ (1 - \tau) \overbrace{L\Big(\mathcal{P}^*\big(\boldsymbol{g}(t_1)\big)\,;\text{NN}_\Theta(\boldsymbol{g}(t_1), t)\Big)}^{\text{loss based on matching domain-knowledge}}$$

that incorporates the effect of any user-provided prior model $\mathcal{P}^*$, using a tuning parameter $\tau$, and the original loss function $L(\boldsymbol{x}, \widehat{\boldsymbol{x}})$. PHOENIX implements $\mathcal{P}^*$ as a simple linear model $\mathcal{P}^*(\boldsymbol{\gamma}) = \boldsymbol{A} \cdot \boldsymbol{\gamma} - \boldsymbol{\gamma}$, where $\boldsymbol{A}$ is the adjacency matrix of likely connectivity structure based on prior domain knowledge (such as experimentally validated interactions) with $\boldsymbol{A}_{ij} \in \{+1, -1, 0\}$ representing an activating, repressive, or no prior interaction, respectively.

For synthetic experiments, we used the simple linear model: $\mathcal{P}^*(\boldsymbol{\gamma}) = \boldsymbol{A}^{\sigma\%} \cdot \boldsymbol{\gamma} - \boldsymbol{\gamma}$, where we chose $\boldsymbol{A}^{\sigma\%}$ to be noisy/corrupted versions of the adjacency matrices of ground truth networks $G_{350}$ and $G_{690}$ (details in B.3.1). We set activating and repressive edges in $\boldsymbol{A}^{\sigma\%}$ to 1, while "no interaction" was represented using 0.

### B.10.4  Algorithm for efficiently retrieving encoded GRN from trained PHOENIX model

We start with PHOENIX's prediction for the local derivative given a gene expression vector $\boldsymbol{g}(t) \in \mathbb{R}^r$ in an $r$-gene system:

$$\widehat{\frac{d\boldsymbol{g}(t)}{dt}} = \text{ReLU}(\boldsymbol{v}) \odot \Big[\boldsymbol{W}_\cup\{\boldsymbol{c}_\Sigma(\boldsymbol{g}(t)) \oplus \boldsymbol{c}_\Pi(\boldsymbol{g}(t))\} - \boldsymbol{g}(t)\Big], \quad \text{where}$$

$$\boldsymbol{c}_\Sigma(\boldsymbol{g}(t)) = \boldsymbol{W}_\Sigma \phi_\Sigma(\boldsymbol{g}(t)) + \boldsymbol{b}_\Sigma \quad \text{and} \quad \boldsymbol{c}_\Pi(\boldsymbol{g}(t)) = \exp \circ(\boldsymbol{W}_\Pi \phi_\Pi(\boldsymbol{g}(t)) + \boldsymbol{b}_\Pi)$$

A trained PHOENIX model encodes interactions *between* genes primarily within the gene-specific multipliers $\boldsymbol{v} \in \mathbb{R}^r$, and the weight parameters from its neural network blocks $\boldsymbol{W}_\Pi, \boldsymbol{W}_\Sigma \in \mathbb{R}^{m \times r}$ and $\boldsymbol{W}_\cup \in \mathbb{R}^{r \times 2m}$. This inspired an efficient means of projecting the estimated dynamical system down to a gene regulatory network (GRN) $\widehat{G_n}$. In particular a matrix $\boldsymbol{D} \in \mathbb{R}^{r \times r}$ is calculated, where $\boldsymbol{D}_{ij}$ approximated the *absolute contribution* of gene $j$ to the derivative of gene $i$'s expression

$$\boldsymbol{D} = \boldsymbol{W}_\cup \begin{bmatrix} \boldsymbol{W}_\Sigma \\ \boldsymbol{W}_\Pi \end{bmatrix}.$$

Gene-specific multipliers $\boldsymbol{v}$ are applied, before adapting the marginal attribution approach described by Hackett *et al.* [21]. This resulted in the **dynamics matrix** $\widetilde{\boldsymbol{D}}$ where $\widetilde{\boldsymbol{D}_{ij}}$ was scaled according to the *relative contribution* of gene $j$ to the rate of change in gene $i$'s expression:

$$\widetilde{\boldsymbol{D}_{ij}} = \frac{\boldsymbol{v}_i \boldsymbol{D}_{ij}}{\sum_{j'=1}^n |\boldsymbol{v}_i \boldsymbol{D}_{ij'}|}.$$

### B.11  Subjecting PHOENIX to DASH pruning

With its simple yet powerful architecture, PHOENIX provides an ideal base setting for applying and testing the merits of the discussed neural network sparsification strategies (Section 2), including DASH. We discuss the DASH implementation here and provide implementation details for other sparsification strategies in Appendix B.9.

For DASH, we follow the steps of the $L = 2$ case in Section 3. Note that in PHOENIX we have $k = r$, since we are modeling how a set of $k$ genes (inputs) affects each others derivatives (i.e. $k$ outputs). We apply the steps from Section 3 on each parallel member of the first $\{\boldsymbol{W}_\Sigma, \boldsymbol{W}_\Pi\}$ and second $\{\boldsymbol{U}_\Sigma, \boldsymbol{U}_\Pi\}$ layers. For the first layer, we leverage prior knowledge of transcription

factor coregulation (often available in the form of protein-protein interaction matrices) to formulate $C \in \mathbb{R}^{k \times k}$. This is then used to calculate pruning scores $\Omega_{\Sigma}, \Omega_{\Pi}$ for $W_{\Sigma}, W_{\Pi}$. Similarly, for $U_{\Sigma}, U_{\Pi}$ in the second layer, we utilize $P \in \mathbb{R}^{r \times k}$ which are easily obtainable motif map matrices encoding prior knowledge of transcription factor binding sites around genes to calculate pruning scores $\Psi_{\Sigma}, \Psi_{\Pi}$.

---

**Algorithm 1** Computing $\Omega_{\Sigma}^{(t)}, \Omega_{\Pi}^{(t)}, \Psi_{\Sigma}^{(t)}, \Psi_{\Pi}^{(t)}$

**Inputs:** weights $W_{\Sigma}^{(t)}, W_{\Pi}^{(t)}, U_{\Sigma}^{(t)}, U_{\Pi}^{(t)}$; priors $C, P$; epoch $t$;
         previous scores $\Omega_{\Sigma}^{(t-1)}, \Omega_{\Pi}^{(t-1)}$; tuning $\lambda_1, \lambda_2$

- Normalize $W_{\Sigma}^{(t)}$ to get $|\widetilde{W_{\Sigma}^{(t)}}|$ (see Appendix B.2)
- Similarly obtain $|\widetilde{W_{\Pi}^{(t)}}|$, $|\widetilde{U_{\Sigma}^{(t)}}|$, and $|\widetilde{U_{\Pi}^{(t)}}|$
- If $t = 0$
    - Initialize $\Omega_{\Sigma}^{(t)}$ and $\Omega_{\Pi}^{(t)}$ randomly with values from a standard Gaussian
- Else
    - $\Omega_{\Sigma}^{(t)} := (1 - \lambda_1)|\widetilde{W_{\Sigma}^{(t)}}| + \lambda_1 \left| \text{PInv}_L \left( \Omega_{\Sigma}^{(t-1)\intercal} \right) \cdot C \right|$
    - $\Omega_{\Pi}^{(t)} := (1 - \lambda_1)|\widetilde{W_{\Pi}^{(t)}}| + \lambda_1 \left| \text{PInv}_L \left( \Omega_{\Pi}^{(t-1)\intercal} \right) \cdot C \right|$
- $\Psi_{\Sigma}^{(t)} := (1 - \lambda_2)|\widetilde{U_{\Sigma}^{(t)}}| + \lambda_2 \left| P \cdot \text{PInv}_R \left( \Omega_{\Sigma}^{(t)} \right) \right|$
- $\Psi_{\Pi}^{(t)} := (1 - \lambda_2)|\widetilde{U_{\Pi}^{(t)}}| + \lambda_2 \left| P \cdot \text{PInv}_R \left( \Omega_{\Pi}^{(t)} \right) \right|$

---

### B.12 Ablation study using a plain MLP instead of PHOENIX as the base model

We believe that DASH should remain performant even when using a base model **that is different from PHOENIX**. To be more explicit, the base model would give a new expression for $\text{NN}_\Theta(g(t), t)$ in B.10.2.

For `PHOENIX` as base model we have: $\text{NN}_\Theta(g(t), t) = \text{ReLU}(v) \odot \left[ c_\cup(g(t)) - g(t) \right]$.

For `alternative base model` we have: $\text{NN}_\Theta(g(t), t) = $ `alternative base model`$(g(t)) - g(t)$

In our ablation experiments, with used a simple **two-layer MLP with ELU activation functions** as the alternative base model . The choice of ELU activation is motivated by the PathReg paper [1]. We chose the number of hidden neurons such that the total number of parameters was comparable between the MLP and the PHOENIX base models. We tested a few sparsification strategies against DASH on this new base model, for both synthetic data and the bone marrow data.

