# OpenReview forum: "Pruning neural network models for gene regulatory dynamics using data and domain knowledge"
_NeurIPS.cc/2024/Conference — NeurIPS 2024 poster_

### Official Review · Reviewer_pC8f · 2024-07-02

**Soundness:** 3
**Presentation:** 3
**Contribution:** 3
**Rating:** 5
**Confidence:** 3

**Summary:**

The authors proposed an interesting model called DASH. The model can recover the gene interact relationships in a sparsity way and make a high accuracy.

**Strengths:**

The authors proposed an interesting method to model the cells.

The insights are straightforward for biology, and the regulatory network should be sparse and accurate.

**Weaknesses:**

The paper is not well organized. E.g., The explanation of the equations and experiments. (see questions)

More biological background should be included for better understanding.

**Questions:**

Can the knowledge be updated by the model?
Can you model and simulate the different activities of the gene regulatory? E.g. activate the genes or deactivate the genes.
How to align the parameter n with \Omiga in DASH for H=2? What is its biological meaning and align with the prior knowledge?
How to measure the accuracy? Do you consider whether the reconstructed relations exist in biology but have not been discovered? E.g. regulatory network rewiring in the cells
How to do the pathway analysis? How to get the enriched pathways?

You use Fig. 1 and Fig. 3 but Figure 2. Please ensure they are in the same form.

**Limitations:**

The authors don't set a limitation section in the paper. The authors proposed a method that simulates the bio system. However, the biology system is more complex. More things can be discussion in the future.

---

> ### Author Rebuttal · Authors · 2024-08-07
>
> **More biological background**: Thank you for the suggestion. With the additional space, we will add further introductory background including a simple figure of how proteins and genes are interacting, and how such an interaction is represented in the prior matrices, and further explanations on the biological mechanism of gene regulation and its impact on disease, in particular cancer, where these regulatory networks are perturbed leading to the particular diseases, which makes GRN analysis so relevant. If you have further specific requests on what needs elucidation, we would appreciate your feedback.
>
> **Can knowledge be updated by the model**: Yes, definitely. While the prior knowledge is a great starting point, DASH sparsifies based on *both* the prior knowledge and the patterns it learns from the data itself. If there are new patterns in the data that are not present in the prior knowledge, then the final model will have learned updated relationships on top of the prior domain knowledge.
>
> **Simulating effect of activating or deactivating genes**: Yes, this is something the PHOENIX model itself (i.e. the base model on which we benchmark the different sparsification strategies) is able to do. We can intervene on an input and set it to zero and analyze the effect on the predicted dynamics.
>
> **Calculating $\Omega$**: We are sorry for the confusion, during a revision of the paper we changed the notation inconsistently. The prior should be $C\in \mathbb{R}^{k\times k},$ where $k$ is the number of genes. Similarly $\boldsymbol{{\Omega^{(t)}_1}^{\intercal} \Omega_1^{(t)}} \in \mathbb{R}^{k\times k}$. So the product of the score matrix reflecting the first layer should align with the prior input-input relationship matrix $\boldsymbol{C}$. Similarly, $\boldsymbol{{\Omega^{(t)}_1} \Omega_2^{(t)}} \in \mathbb{R}^{r\times k}$ should align the prior output-input relationship matrix $\boldsymbol{P}$. The score is based on a basic reasoning that also underlies GRN inference techniques like [1], which argues that proxies of TF-TF interactions ($P$) or gene-gene interactions ($C$) are most readily available. The DASH pruning score aligns these proxies $P$ and $C$ with corresponding functions of the edges in the neural network. We have also provided more mathematical motivation below to Reviewer #4.
>
> **Finding new/undiscovered biological relations**: Such a finding would be ideal and our model has been designed for this purpose.
> Inspired by your and other reviewer's question, we further analysed the obtained networks. Focusing on the Heme signaling pathway, which was uniquely identified by our suggested approaches. It turns out that this signaling pathway is highly relevant in breast cancer, we then proceeded to extract interaction partners from our model, i.e., the factors most highly affecting the Heme signaling molecules' dynamics, that could potentially be used as a new drug target. We provide further information on this finding in the general rebuttal response. We hope to discover further relations in the future, when we apply this approach in the field.
>
>
>
> **Pathway analysis**: As described in Appendix B8,  we first take a trained and sparsified model and compute gene influence scores, which we then use to compute pathway influence scores via permutation tests. The $z$-scores of the permutation tests are a measure of pathway enrichment.
>
> **Labelling consistency** : Thank you for pointing this out. We will ensure consistency in a revised version.
>
> [1] D Weighill et al. Gene Regulatory Network Inference as Relaxed Graph Matching, AAAI, 2021

---

> > ### Comment · Reviewer_pC8f · 2024-08-13
> >
> > Thanks for your response. For the biological background, I think it is more important to describe what is the meaning of some biological names. For example, what is a regulatory factor? You also give the transcription factors as one example of regular factors, which is harder to understand. I believe the biological background can provide a more easy way for reviewers to understand the importance of your solutions for the problems. The paper needs further polishing for publication. I will keep my score.

---

> > > ### Author Response · Authors · 2024-08-14
> > > **Adding biological background**
> > >
> > > We would be happy to include further explanations and we do believe this would make for a simple revision -- after all the motivation, method, and experiments are almost the same, only additional background of the particular application is added, which is usually considered a minor change.

---

> > > > ### Author Response · Authors · 2024-08-14
> > > >
> > > > Thank you for the useful suggestion. It is important to note that a great deal of the motivation for this work is the need in the biological sciences to have methods that balance predictive power with explainability. A highly predictive model has some utility but we want to understand why the system evolves from State A to State B so that we can search for meaningful ways to perturb the system in its evolution and affect the outcome. For example, if we have a tumor that is responsive to chemotherapy, we would want to block the tumor from evolving to a chemo-resistant state. A model with high predictability and high explainability would allow us to develop hypotheses about interventions. We would be happy to further clarify the motivation and to provide more insight into the biology of the systems we are modeling.

---

### Official Review · Reviewer_QJVK · 2024-07-11

**Soundness:** 2
**Presentation:** 2
**Contribution:** 2
**Rating:** 3
**Confidence:** 4

**Summary:**

The authors propose a network pruning approach which is guided by prior biological domain knowledge, which they call Domain-Aware Sparsity Heuristic (DASH). They aim to obtain highly sparse networks, which align with known biology and gene regulatory dynamics. To do so they propose computing pruning scores which combine learned weights with prior domain knowledge, balanced by a tunable parameter $\lambda$. The authors apply DASH to PHOENIX, a neuralODE designed to model gene regulatory dynamics and conduct experiments on four datasets, one synthetic and three real world. They also compare DASH to other existing pruning techniques, such as $L_0$, PathReg and SparseFlow. The experiments compare DASH to these techniques in terms of sparsity, accuracy in recovering known biological relationships and MSE of predicted gene expression values, as well as pathway analysis.

**Strengths:**

The problem addressed in this paper is scientifically very relevant and topical. How to extract real biological insight from gene expression information, in particular as it pertains to gene regulatory networks in an open and important problem. Using domain knowledge priors to constrain the problem and guide pruning of a neural network is a logical approach to this question. While using prior domain knowledge to regularise a neural network is a well studied area, this paper focuses on how to introduce these constraints to continuous-depth NeuralODE models. The authors conduct extensive experiments on both synthetic and real-world datasets, as well as comparing against other well established pruning methods. I think this is relevant and important research, but in my opinion there are four main issues to address in this paper: the mathematical formulation for DASH is very confusing, there is an over emphasis of results obtained on synthetic data, as well as a reliance on a metric (balanced accuracy) which might potentially lead to inflating results, and a lack of clarity about potential confounding effects between DASH/PHOENIX. I will detail each of these concerns below.

**Weaknesses:**

I think this is relevant and important research, but in my opinion there are four main issues to address in this paper: the mathematical formulation for DASH is very confusing, there is an over emphasis of results obtained on synthetic data, as well as a reliance on a metric (balanced accuracy) which might potentially lead to inflating results, and a lack of clarity about potential confounding effects between DASH/PHOENIX. I will detail each of these concerns below.

1 - Mathematical formulation of DASH, in the case of L=2 (at the top of p.5 - I have no numbering on the PDF here for some reason):

  - In line 139 you define your weight matrix as  $W \in \mathbb{R}^{k \times r}$, where $k$ is your input corresponding to your genes - yet a paragraph later you transpose your notation and start using $W_1\in \mathbb{R}^{m\times k}$, $W_2\in \mathbb{R}^{r\times m}$, which is confusing.
   - You then define an input-input matrix $C \in \mathbb{R}^{n\times n}$ which should be $\mathbb{R}^{k\times k}$ as you defined your inputs a few lines above, you follow by saying that $W_1$ has $n$ inputs and that the matrix product $\Omega_1^{(t)^T} \Omega_1^{(t)}  \in \mathbb{R}^{n\times n}$ -  this doesn't make sense dimensionally.
   - You posit $\Omega_1^{(t)^T} \Omega_1^{(t)}  \in \mathbb{R}^{n\times n}$  approximates to $C$ without first clearly defining what $\Omega_1^{(t)}$ is and without clearly enunciating why this is most likely a good approximation.
   - You then define a recurrent relation between $\Omega_1^{(t)}$ and $\Omega_1^{(t-1)}$, which needs an initial condition to be well defined. Furthermore, this temporal dependency is not present in the L=1 case.
   - Why are you using the left and right pseudoinverses on $\Omega_1^{(t-1)}$ and $\Omega_1^t$ respectively? It would be good to take time to explain this approach in more detail, as this is the central premise of this paper. You don't need to define what the left and right pseudoinverse of a matrix is. You do need to explain why you are using them here and to what end.
   - The product of $W_2^{(t)} \cdot W_1^{(t)}$ should be $\in \mathbb{R}^{k\times r}$, not $\mathbb{R}^{o\times n}$


The definition of DASH is the central premise of the paper, yet there are many notation errors, variables which are not properly defined, and most importantly only vaguely motivates why the proposed substitution makes sense. I suggest you introduce all the notation you will use, even if it seems trivial, and be more careful with errors in notation. It would be great if you expanded on this paragraph by giving more intuitions behind the physical reasoning, as well as explaining the mathematical steps described.

2 - Over emphasis of results obtained on synthetic data:

A significant portion of results (Table 1, Figure 2 and 3) and discussion focus on synthetic data. Although useful for initial validation and exploration, an in depth examination of the three real-world datasets studied would be more relevant. The data generation process likely oversimplifies the complexities of real regulatory networks, which could lead to an unrealistic assessment of the model's performance.  The results obtained on the real-world datasets are not as compelling as those obtained on the synthetic dataset suggesting DASH is not generalising as well as it could and might be fitting to characteristics of the synthetic dataset. Expanding on the results obtained for the real-world datasets would help address these concerns. For example, the results presented in A.3 Table 6 for the bone marrow data differ significantly from the Sparisity and MSE results obtained for $L_0$, C-NODE and PathReg in the PathReg paper. I would be interested to know more about these differences. Could it be coming from using PHOENIX as a baseline, rather than the NN model used in PathReg?  In my opinion the results on the bone marrow dataset should be in the main body of the work, as it reproduces an experiment from the PathReg paper, permitting a more direct comparison between the methods.

3/4 - Balanced accuracy to validate biological alignment \& confounding effects between DASH/PHOENIX

The balanced accuracy for biological alignment metric is not clearly defined in the main body of the article, yet is one of the main results in Table 1, 2 and 6, where the authors argue that because DASH is obtaining higher balanced accuracy scores it is better able to recover true biology. They point towards B.5, where they say "To validate biological alignment of trained and sparsified models, we extracted GRNs from each models (as explained in B.10.4), and compared back to the validation networks." In B.10.4 they describe retrieving GRN from the PHOENIX model. This lack of clarity regarding how the biological alignment is measured makes it difficult to evaluate these results. Is it possible that the extraction process itself could be biased towards producing results that align with the prior knowledge used in DASH? This could artificially inflate the balanced accuracy scores for DASH and BioPrune. Could it be there is a confounding effect in the ablation study coming from applying the sparsification methods only to the PHOENIX model? To check this the results would need to be expanded to include a different SOTA neuralODE model to verify DASH works across different models and is not a result of the DASH/PHOENIX combination.

**Questions:**

In addition to the questions and concerns I raised above, I have the following questions/comments:

- For clarity, you apply all the pruning  methods mentioned - $L_0$, C-NODE, PathReg, DST, IMP, SynFlow, SparseFlow - to PHOENIX?

- What's the difference between PHOENIX with biological regularisation and BioPrune?

- In Results, you say PINN+MP and BioPrune are your models, yet they both correspond to post-hoc pruning if I understand correctly. Therefore, neither of them have undergone pruning with DASH. Yet, both perform strongly, with PINN+MP outperforming DASH on MSE in Table 1 and seemingly doing a better job at reconstructing ground truth relationships in Figure 3. Why do you think that is?

- The main difference between models in Table 2 seems to be the Balanced Accuracy, but is it not expected for DASH to recover similar levels of sparsity to BioPrune given you are "forcing" convergence towards $C$ in your model and hence achieving a higher Balanced Accuracy?

- If one of the takeaways of DASH is the reduced training time and/or memory usage it would be good to see training/inference time or GFLOPs.

- Can this be used to generate new biological knowledge? Some discussion on this topic would enrich the conclusions of this paper in my opinion.

**Limitations:**

I do not think the potential limitations of DASH are adequately discussed, see my comments above.

---

> ### Author Rebuttal · Authors · 2024-08-07
>
> **Mathematical formulation**:  We apologize for the typos in dimensionality. Below we provide a brief note that includes all the corrected notation, definitions of key variables, and motivations behind the mathematical steps, especially pseudo-inverses.
>
> *DASH for $L=1$.*
> For a single layer NN, with $k$ input and $r$ output neurons and corresponding weight matrix $\boldsymbol{W}$ should be $\in \mathbb{R}^{r\times k}$, and we compute pruning scores $\boldsymbol{\Omega} \in \mathbb{R}^{r\times k}$ using domain knowledge $\boldsymbol{P} \in \mathbb{R}^{r\times k}$. At any epoch $t$, we simply use $\boldsymbol{P}$ as the prior-based portion of $\boldsymbol{\Omega^{(t)}}$, the pruning scores at that epoch.
>
> *DASH for $L=2$.*
> We are sorry for the confusion, during a revision of the paper we changed the notation inconsistently. The dimensionalities you provided are correct. The prior should be $C\in \mathbb{R}^{k\times k},$ where $k$ is the number of genes. Similarly $\boldsymbol{{\Omega^{(t)}_1}^{\intercal} \Omega_1^{(t)}} \in \mathbb{R}^{k\times k}$. Now, since $\boldsymbol{W_1^{(t)}}\in \mathbb{R}^{m\times k}$ learns how the $k$ inputs are encoded by $m$ hidden neurons, we surmise that the matrix product $\boldsymbol{{\Omega^{(t)}_1}^{\intercal} \Omega_1^{(t)}} \in \mathbb{R}^{k\times k}$ should approximately align with our prior knowledge of how the inputs co-vary, i.e. with $\boldsymbol{C}$. Since solving $\boldsymbol{{\Omega^{(t)}_1}^{\intercal} \Omega_1^{(t)}} = C$ is not feasible, we initialize $\boldsymbol{\Omega_1^{(0)}}$ as all 1s, and solve  $\boldsymbol{{\Omega^{(t-1)}_1}^{\intercal} \Omega_1^{(t)}} = C$ instead. This is what motivates using the pseudoinverse.
>
> **Synthetic vs real world results**: We agree that the synthetic data is only one part of the story, yet, it highlights distinct differences between the methods in a setting with available ground truth. If methods already fail to perform well on this "oversimplified" data, or show specific biases in terms of recovered structure, then such experiments are valuable. Moreover, we disagree that the real world results are "not as compelling", we get close to 90\% accuracy in network recovery, which is much better than existing work (see Table 2). Naturally, there is a sparsity difference to the PathReg paper as the neural network architecture is different, thus different sparsity levels are required, as you correctly pointed out. We do recover a ranking of methods similar to the PathReg paper, which speaks for consistency. We are happy to dedicate more space to bone marrow results with the additional page in case of acceptance.
>
> **Balanced accuracy**: Thank you for pointing that out, we will make this more concise and available in the beginning of the experiment section. In brief, the balanced accuracy measures whether an edge is correctly reconstructed weighted by the sparsity of the ground truth graph. The extraction is the same for all models, as each use the base PHOENIX architecture. The metric is computed with respect to a ground truth (in case of synthetic data) or gold standard (in case of real data). The gold standard ChIP-seq experiments are *not* included in and are distinct from the prior.
>
> **DASH+PHOENIX confounding**: We believe that DASH should remain performant even when using an base model that is different from the PHOENIX. Hence we performed additional experiments (Table 1 in rebuttal PDF), with a simple two-layer MLP with ELU activation function as base model. The choice of ELU activation is motivated by the PathReg paper. We test a few sparsification strategies on this new base model applied to both synthetic data and the bone marrow data. We found DASH to still be performant.
>
> **Questions about PHOENIX**: We apply all the sparsification strategies to the PHOENIX *base model*, that is the PHOENIX model without the biological regularization, a two-layer MLP with activation functions that resemble Hill kinetics. BioPrune is a pruning-based strategy that uses prior knowledge of the GRN to *explicitly* sparsify the neural network by setting weights to zero. Biological regularization takes an indirect route and *implicitly* encourages the neural network parameters towards zero through a penalty-term in the regularizer.
>
> **PINN+MP vs DASH**: PINN+MP and BioPrune are novel models, as they have not been proposed in this context and we primarily discuss them as baselines compared to DASH.  On synthetic data, PINN+MP is indeed a strong contender, with an MSE *comparable* to DASH (within error margin), yet deliver much less accurate representation of the underlying gene regulatory system (cf. balanced accuracy). Regarding Fig 3, the difference is barely visible in the small image. We provide a high-res version (Fig 1 in rebuttal PDF) where we also display the error between the inferred and true relationship. DASH outperforms PINN+MP which recovers lots of spurious structure.
>
> **Table 2**: Yes, DASH and BioPrune do encourage the alignment of the model with the *prior knowledge*. But, the balanced accuracy is measured by comparing the sparsified model to ChipSeq *validation data* which is experimentally independent of the priors in question.
>
> **Training time**: This is not a primary focus of our work, as runtime is generally negligible in comparison with applications in vision or language, e.g. DASH on PHOENIX takes approximately 40 minutes on the breast cancer data. The main motivation for pruning here is interpretability and alignment of sparsified models with known biology.
>
> **New knowledge**: In principle, this approach can be used to generate new knowledge. We here focused on insights we can validate, which naturally means known knowledge. As a response to the question, we added a finding on Heme signaling based on new insights and its potential role in therapeutic design, which we will also add to the discussion. Due to the character limit, please refer to the general rebuttal for the details of this finding.

---

> > ### Comment · Reviewer_QJVK · 2024-08-12
> > **Response to Authors Rebuttal**
> >
> > I thank the authors for the time and effort put into their rebuttal, especially adding the ablation results comparing PHOENIX to another baseline model in Table 1 (R) for bone marrow data and simulated yeast time-series data. I also appreciate the authors addressing the mathematical notation errors in the text.
> >
> > However, I still do not understand the discrepancy between obtained results and those shown in PathReg (cf. Appendix C where they show extended results, where the sparsity levels and MSE obtained differ markedly from those reported here). Moreover, I strongly feel the overall structure of the paper needs substantial reorganising and editing to make it more readable and accessible, with an emphasis on properly motivating the DASH model both mathematically and biologically, introducing PHOENIX early on and clearly enunciating the difference between PINN+MP/BioPrune and DASH. Given this would require substantial modification of the work as it stands, I am maintaining my original rating.

---

> > > ### Author Response · Authors · 2024-08-14
> > >
> > > We are happy that you appreciate the additional results. Note that the results are not directly comparable as the number of layers is different. Our main goal is to show that our method is also applicable to other architectures (MLP + ELU activations as in the pathreg paper) rather than replication of previous results. Regarding the requested changes, we do believe that the suggested changes about a restructuring of text is usually considered a minor change as motivation, method, as well as experiments are still similar.

---

### Official Review · Reviewer_m1mG · 2024-07-12

**Soundness:** 2
**Presentation:** 3
**Contribution:** 3
**Rating:** 4
**Confidence:** 4

**Summary:**

Gene regulatory network inference is an important, but difficult problem.

The manuscript explores a novel approach to build domain knowledge into a general NODE model for this problem via pruning. The approach could work in other areas.

**Strengths:**

Gene regulatory network inference is an important, but difficult problem.

The manuscript explores a novel approach to build domain knowledge into a general NODE model for this problem via pruning. The approach could work in other areas.

**Weaknesses:**

It’s very hard to match real data biases for the prior GRN when studying simulated data. I’m struggling to understand whether the noise % refers to this GRN prior or the expression data. 5% noise for the prior seems completely unrealistic and would render these experiments irrelevant to real data analysis.

Analyzing real gene expression data, the authors use priors based on TF motifs mapped to promoter sequences, which enable DASH to identify genes with true TF ChIP binding with apparently high accuracy. Regulatory network inference method evaluations generally find that genes with nearby TF binding events do not correspond well to genes whose expression changes upon TF perturbation. Evaluation with TF perturbation data is the gold standard here and would make for a more compelling DASH evaluation.

**Questions:**

In Fig 2, what does it mean for DASH to have 2x fewer parameters than BioPrune but achieve greater GRN accuracy? Isn’t BioPrune simply using the GRN?

In Fig 2, what is ”Base model”?

It would be helpful to understand why PHOENIX with the same GRN prior information, but using regularization instead of pruning, falls behind the DASH pruning strategy.

I’m not really sure what I’m supposed to be able to understand from Figure 3. I can’t see any of the specific matrix entries well enough to compare them.

**Limitations:**

Yes

---

> ### Author Rebuttal · Authors · 2024-08-07
>
> **Noise levels**: The noise was applied to both the expression data as well as the GRN prior, which we further describe in Appendix B1 and B3. We tested on three different noise levels (0\%, 5\%, 10\%). We included results from the 5\% setting in the main paper and the results for the remaining noise levels are in A1.
> Generally, we find that DASH is more robust to noise than its competitors and the performance gap increases with the noise level.
> This insight is also supported by the fact that DASH identifies more meaningful biology on real world data, which is indeed very noisy.
>
> **Evaluation with TF perturbation data**: This is a great observation and we fully agree. However, this is prohibitively expensive and difficult to generate for time-course data and is hence not available up to our knowledge, especially on the genome-scale ($>10^4$ genes). We thus resorted to the next best thing, which is TF ChIP-seq data. If we could obtain perturbation data in the future, we would be eager to extend DASH to it.
>
> **DASH vs BioPrune**: It is correct that BioPrune is simply using the prior GRN to calculate pruning scores, unlike DASH which takes *both* the prior GRN and the learned neural network weights into account when calculating pruning scores.
> However, the achieved sparsity is determined by the validation set. Thus, BioPrune can still contain edges that are not in the prior.
> The pruning process in concretely described in Appendix B.2.5, where we describe how each training epoch consisted of the entire training set being fed to the model, preceded by any pruning step that is prescribed by the pruning schedule. Training is terminated if the validation set performance fails to improve in 40 consecutive epochs. Upon training termination, we have obtained a model that has been iteratively sparsified to an extent that fails to improve the validation set performance. We discovered in our experiments that DASH's combination of data-based information (from the learned weights) and prior information (from the prior domain knowledge or GRN), leads to this achieved sparsity being much better than that of BioPrune. When we compare the GRN encoded by this sparse model structure back to the ground truth GRN, we see that the sparser model obtained by DASH encodes a much more accurate GRN than the denser model obtained by BioPrune.
>
> **Base model**: This is the PHOENIX base architecture without any form of prior regularization. It is a fully connected two layer MLP  with activation functions resembling Hill-like kinetics and  gene specific multipliers for improved trainability. This base model is then subjected to the different sparsification strategies being benchmarked.
>
> **Why prior-informed regularization falls behind prior-informed pruning**: This is because a pruning-based strategy (such as DASH) *explicitly* sparsifies the neural network by setting weights to zero, while regularization takes an indirect route and *implicitly* encourages the neural network parameters towards zero. The former leads to better sparsification in pruning based strategies, which is similar to the traditional pruning literature, where explicit pruning such as IMP, SNIP, SynFlow, etc usually outperform implicit $L_p$-based approaches.
>
> **Regarding Figure 3**: The figure aids a qualitative comparison, where the overall structure of the matrix recovered by each method should be compared to the ground truth matrix (leftmost). It becomes evident that DASH as well as PINN+MP are much more aligned with the ground truth. A good quantitative measure of alignment here is the mean squared error (MSE) between the inferred and the ground truth relationships. We calculate this and include it in Additional Figure 1 (see rebuttal PDF), which clearly shows that DASH outperforms PINN+MP.  We truncated Additional Figure 1 to only DASH and PINN+MP and increased the contrast to visualize the difference between these two, as they can appear to be very similar in Figure 3 of the paper. We will add this quantitative measure to all sparsification strategies in Figure 3 in the final version if the paper is accepted.

---

> > ### Comment · Reviewer_m1mG · 2024-08-11
> >
> > I have read the authors’ rebuttal. I hope you’ll include these clarifications and additional information in your revision.
> >
> > First, I apologize that I missed copy pasted my summary of your paper. Here was the original version I wrote in my own notes:
> > *This paper proposes a new method called DASH (Domain-Aware Sparsity Heuristic) for pruning neural ordinary differential equation (NODE) models to infer gene regulatory dynamics from time series gene expression data. The key innovation is incorporating prior biological knowledge as soft constraints during pruning to achieve more interpretable and biologically plausible sparse models. The authors evaluate DASH against existing pruning methods on both synthetic and real gene expression datasets, showing it can achieve high sparsity while maintaining predictive accuracy and recovering “known” regulatory interactions. Overall, the paper demonstrates value from biology-informed pruning for inferring interpretable gene regulatory networks from dynamic data.*
> >
> > I don't understand why you can't use TF perturbation data to evaluate your method. There's a large literature on this. E.g. see Kamal, A. et al. GRaNIE and GRaNPA: inference and evaluation of enhancer‐mediated gene regulatory networks. Mol Syst Biol e11627 (2023).
> >
> > I remain concerned that the simulations use impractically small noise levels on the prior.
> >
> > Altogether, I’ll maintain my current scores.

---

> ### Author Response · Authors · 2024-08-12
> **Clarification ChIP-seq and perturbation experiments with additional experiments**
>
> Thank you for staying engaged in the discussion!
>
> As we tried to convey in our rebuttal, in principle, if large-scale perturbation data (e.g., by perturb-seq experiments) would exist for the tissues of interest such as breast cancer tissue, we would be happy to use it. But such experiments are usually (1) prohibitively expensive and (2) can not measure interventional perturbation effects within the tissue, but only on a cell-level - the tissue has to be extracted for gene-level perturbation experiments as they are *in-vitro*. Hence, we lose the ability of determining true effects in the whole tissue when using a perturb-seq-like experiment, while ChIP-seq gives a snapshot of the actual tissue state.
>
> There also seems to be a misunderstanding, the provided reference [1] talks about expression difference after a *global* perturbation, for example an infection of cells. This gives expression difference of individual genes, but not how this was affected by other genes. In particular, this does not measure "causal" gene--gene effects such as the famous perturb-seq experiments based on CRISPR, which could be used as gold standard ground truth.
>
> Considering perturb-seq and assuming it exists for our data, there is also an inherent limitation using (gene expression) perturbation data: we might see which genes $X$ have a causal effect (strength of up- or down-regulation) on a particular other gene $Y$, but these could be  secondary or tertiary effects of the form $X \rightarrow Z_1 \rightarrow \ldots \rightarrow Y$, which should not correspond to links in the GRN. Only the immediate parent in the causal graph would ideally be represented there. With ChIP-seq, we do get the direct causal relationship (a TF $X$ *is binding* to the promoter of a gene $Y$), but lose the ability to tell what the exact causal effect is (up- or down-regulation, how strong, etc.).
>
> That being said, in response to your review, we considered a comparison to a tf-inducing experiment (i.e., a bio-engineering to induce a particular TF's expression) on yeast, which was then manually curated to derive a "true" causal GRN [2], i.e., aiming to remove secondary and tertiary effects as discussed above. We note that while the underlying organism is the same yeast, the data was derived in different conditions, hence we expect differences in the GRNs and focus on *relative* performance differences between methods. To summarize the results, which we provide in the table below, the ranking of methods is similar to the comparison to the ChIP-seq gold standard, and there is still a large ($>10$ percentage points) improvement of DASH over existing state-of-the-art methods.
>
> | Strategy | Sparsity | Bal. Acc. (ChIP-seq) | Bal. Acc. (TF Perturb.) |
> | :------- | :------: | :-----------------: | :----------------------: |
> | None/Baseline   | 0.10\% | 49.87\% | 49.92\% |
> | $L_0$   | 34.43\% | 48.43\% | 49.28\% |
> | C-NODE   | 10.89\% | 50.04\% | 50.17\% |
> | PathReg   | 12.09\% | 50.11\% | 49.92\% |
> | PINN   | 0.17\% | 49.93\% | 50.01\% |
> | DST  | 77.80\% | 49.92\% | 50.33\% |
> | IMP| 83.22\% | 49.99\%  | 48.45\% |
> |Iter. SynFlow  | 85.65\% | 49.57\% | 49.77\% |
> | SparseFlow  | 95.22\% | 49.89\% | 51.58\% |
> | BioPrune   | 94.69\% | 79.23\% | 64.50\% |
> | DASH | **97.18\%** | **88.43\%** | **66.79\%**|
> | PINN + MP  | 95.01\% | 55.39\%  | 52.95\% |
>
> We would be happy to include further comparison in the future and appreciate any direct pointer to gene-level perturbation data relevant to us. Additional noise experiments will be provided in a separate comment.
>
> [1] Kamal, A. et al. GRaNIE and GRaNPA: inference and evaluation of enhancer‐mediated gene regulatory networks. Mol Syst Biol e11627 (2023).
>
> [2] Hacket, SR. et al. Learning causal networks using inducible transcription factors and transcriptome‐wide time series. Mol Sys Bio 16: e9174 (2020).

---

> ### Author Response · Authors · 2024-08-12
> **Additional experiments with more noise in prior**
>
> We understand your concern and added additional experiments with 20\% respectively 40\% noise on the prior, given in the table below. As expected, we do see a slight decrease in performance for prior-based methods correlated with the increase in prior noise. Yet, DASH still outperforms all existing work in terms of GRN accuracy even for 40\% noise in the prior.
> We will add this additional analysis to the manuscript to provide a better discussion of robustness to prior noise.
> We thank the reviewer for their constructive feedback.
>
>
> | Strategy | Prior corruption | Sparsity(\%) | Bal. Acc.(\%) | MSE ($10^{-3}$)  |
> | :----- | :---: | :---: | :---: | :---: |
> | None/Baseline | - | 11.5 | 54.8 | 3.6 |
> | $L_0$ | -  | 34.7 | 61.3 | 6.1 |
> | C-NODE | -  | 10.7 | 60.5 | 1.9  |
> | PathReg | -  | 59.7 | 64.2| 6.1 |
> | DST |- | 94.3 | 72.3 | 4.2 |
> | IMP | - | 86.1 | 63.2 | 4.1  |
> | Iter. SynFlow | - | 79.1 | 60.0 | 2.3 |
> | SparseFlow | - | 95.8 | 72.8 | 2.9 |
> | PINN |  0\%  | 11.3 | 60.3 | 2.3  |
> | PINN |   20\%  | 12.4 | 60.8 | 3.1 |
> | PINN |   40\%  | 11.2 | 60.6 | 2.7 |
> | BioPrune |  0\%  | 83.5 | 88.0 | 3.6 |
> | BioPrune|   20\%  | 80.9 | 81.5 | 7.6 |
> | BioPrune |   40\% | 86.8 | 80.1 | 11.1 |
> | DASH |  0\%  | 92.6 | 91.1 | 1.9  |
> | DASH |   20\%  | 92.4 | 86.2 | 6.7 |
> | DASH |   40\%  | 85.9 | 79.5 | 6.1 |

---

### Official Review · Reviewer_YHXc · 2024-07-13

**Soundness:** 3
**Presentation:** 4
**Contribution:** 2
**Rating:** 5
**Confidence:** 4

**Summary:**

The proposed DASH method underscores the importance of interpretability in network pruning for biological discoveries, emphasizing the need for alignment with domain knowledge. Using both synthetic and real data, DASH demonstrates superior performance beyond baselines and offers insights into biological systems.

**Strengths:**

The DASH’s ability to integrate domain-specific information makes the resulting models more robust to noise which is usually a challenge in complex biological data.

**Weaknesses:**

Depending on the size and complexity of the domain knowledge, DASH might be hard to apply to large-scale or highly complex networks.

**Questions:**

1. In Figure 3, what is the percentage of true relationships comparing DASH and PINN + MP?
2. It’s not surprising that incorporating the known domain knowledge enhance the interpretability and learned more meaningful dynamics. Moreover, how does the learned gene regulatory dynamics itself can benefit the downstream disease related outcome predictions?
3. The paper mentioned that DASH has the better quality of inferred (new) knowledge. Is there any discussion or literatures to support the inferred “new” knowledge?
4. How about making lambda a learnable parameter to mimic the complex and dynamic biological system?

**Limitations:**

The authors clearly addressed the limitations.

---

> ### Author Rebuttal · Authors · 2024-08-07
>
> **Large scale networks**: For our particular domain, this problem was addressed by the design of the PHOENIX architecture, which scales up to large and complex networks commensurate to the whole human genome (on the order of $10^4$ genes/dimensions). Here we demonstrate that DASH works effectively on the PHOENIX architecture applied to such large scale datasets (e.g. the breast cancer dataset covers 11165 genes).
>
> **Regarding Figure 3**: For the noise levels of 0%, 5%, and 10%, the percentage of true relationships captured by DASH are 98\%, 95\%, and 88\% respectively, while the corresponding numbers for PINN+MP are 98\%, 98\% and 90\%. However, this is not a good quantitative measure of the how well each method recovers the ground truth, since PINN+MP recovers a lot of spurious features. This is shown in additional Figure 1 (see rebuttal PDF) where we zoomed in to just these two methods and increased the contrast.  A better quantitative measure of alignment is the mean squared error (MSE) between the inferred and the ground truth relationships. We calculate this and include it in additional Figure 1, which clearly shows that DASH outperforms PINN+MP.
>
> **Integration of domain knowledge for downstream disease related outcome tasks**: We agree that incorporating domain knowledge leads to more meaningful learned dynamics, which is why we anticipate our approach DASH to find many relevant practical applications. In Genomics, also in the biomedical domain, the goal is often not *prediction* but *inference*, which is also the case here: We would like to learn more about the inherent structure of the disease in terms of the complex gene regulations. By learning about these structures, such as learning about specific genes that strongly affect other genes in a specific cancer, we can better understand the disease process and ultimately can design better treatments. More concretely, we can for example try to understand the effect of a drug on the gene regulatory network by investigating the change in gene regulatory dynamics between cells exposed to the drug versus those that were not exposed, which is what the authors of PHOENIX -- the model we use as basis for our experiments -- did in their original work. We also added an additional result, where we found an interesting biological insight, which we provide in the general rebuttal response due to character limit. In particular, we show how the insights derived from the GRN can be used to generate a potential new therapeutic approach.
>
>
>
> **"New" knowledge from DASH**: For both breast cancer as well as yeast cell cycle, we use gold standard ChIP-seq data (a biological experiment to locate binding of TFs to the genome) to validate the biology of the inferred GRN. I.e., ChIP-seq experiments on the particular cells gives us a gold standard GRN for validation and we additionally note that this data has not been used for the construction of the prior. For higher-level knowledge, we have included a biological pathway analysis and found that the most relevant pathways identified in our model correspond to key paths in breast cancer progression, yeast cell cycle progression, or, respectively, bone marrow hematopoesis that align with the known biology of these processes.  More concretely, we provide the example of the following pathways: "... *TP53 activity* and *FOXO267 mediated cell death*, both of which are highly relevant in cancer [ 37 , 24 ]". In response to your question, we further investigated the discovered biology focusing on the Heme signaling pathway, which was uniquely identified by our method. We provide more details on these interesting findings in the general author rebuttal due to space constraints.
>
> **A learnable $\lambda$**: Here $\lambda$ is a hyperparameter of DASH. As mentioned in Appendix B4, the $\lambda$ values are determined using a $K$-fold cross validation approach. Specifically, we have automated a process that fits multiple models to the same dataset each with a different set of lambda values chosen from a grid. This automated grid-search is used to optimize the $\lambda$ values, based on predicted MSE on the validation set. This means that $\lambda$ adapts to the complexity of the data. Learning $\lambda$ in a differentiable manner instead would involve differentiating a complicated loss that is defined on cross-validation data, which is usually not efficient.

---

### Official Review · Reviewer_GCM5 · 2024-07-15

**Soundness:** 3
**Presentation:** 2
**Contribution:** 2
**Rating:** 5
**Confidence:** 4

**Summary:**

The paper presents DASH (Domain-Aware Sparsity Heuristic), a new framework for pruning neural network models by incorporating domain-specific knowledge. The primary goal is to improve the interpretability and biological relevance of models used for gene regulatory network (GRN) inference. Traditional pruning methods often fail to reflect biologically meaningful structures, leading to less interpretable models.

### Key Contributions:
- Introduction of DASH: A pruning method that uses both learned weights and prior domain knowledge to iteratively score and prune network parameters, ensuring the resulting models are both sparse and biologically meaningful.
- Improved Model Interpretability: By guiding pruning with structural information about gene interactions, DASH produces models that align better with biological insights compared to traditional methods.
- Experimental Validation:
On synthetic data, DASH outperformed general pruning methods in accurately recovering the underlying GRN.
On real-world gene expression data, DASH identified biologically relevant pathways that other methods missed.
- Robustness to Noise: The framework maintained model robustness even in noisy data environments, showcasing its practical utility.

**Strengths:**

### Originality
The paper introduces DASH (Domain-Aware Sparsity Heuristic), which uniquely integrates domain-specific knowledge into neural network pruning. This approach is innovative as it combines traditional pruning methods with biological insights to enhance model interpretability, a critical need in scientific research.

### Quality
The methodology is robust, with thorough experimentation on both synthetic and real-world datasets. The authors present a detailed comparison with existing pruning methods, demonstrating DASH's superior performance in recovering biologically meaningful structures. The experiments are well-documented, ensuring reproducibility.

### Clarity
The paper is well-organized, with clear explanations of the proposed method and its implementation. The use of visual aids such as figures and tables helps in understanding the results and the effectiveness of DASH. However, simplifying some of the technical details could further enhance accessibility.

### Significance
The contributions of this paper are significant, especially for computational biology. By improving the interpretability of neural networks in gene regulatory dynamics, DASH provides valuable insights that can aid in understanding complex biological processes. This approach has the potential to be applied to other domains, making it a versatile tool for scientific research.

These strengths highlight the paper's contribution to advancing the field of neural network pruning by incorporating domain-specific knowledge, leading to more interpretable and meaningful models.

**Weaknesses:**

### Testing Set Size
One notable limitation of the paper is the relatively small size of the testing set, which comprises only 6% of the total data. This raises concerns about the generalizability and robustness of the reported results. A small testing set can lead to overfitting and may not adequately capture the model's performance on unseen data, especially in cases where the performance differences between methods are quite small.

### Complexity of the Methodology
While the methodology is detailed and thorough, it may appear overly complex for some readers. Simplifying certain aspects or providing more intuitive explanations could improve accessibility and comprehension.

### Biological Validation
The paper could further substantiate the claim that DASH improves biological relevance by involving domain experts and providing concrete examples.

Suggestions for Improvement:

- **Expert Validation**: Include validation from domain experts to verify the biological significance of the results.
- **Concrete Examples**: Provide specific examples where DASH's results align with known biological mechanisms or lead to new hypotheses.

**Questions:**

### 1. Format and Numerical Examples of $P$:
It appears that using prior knowledge alone (i.e., the "BioPrune" baseline) can perform reasonably well for both synthetic and real datasets. However, the description of the domain knowledge used is not entirely clear. It seems that different sets of domain knowledge are employed for different datasets. While we understand the generic form of $P$, can you provide numerical examples for both synthetic and real datasets to illustrate the exact format of $P$?

### 2. Parameter Tuning
How are the lambda values $\lambda$ determined for each dataset? Are they manually tuned, or is there an automated process for selecting these values? Can you provide the results and details on the validation process for tuning these parameters?

### 3. Generalizability (to other neural network, to other domain)
The paper demonstrates the effectiveness of DASH on gene regulatory networks. How generalizable is this approach to other domains or types of neural networks (CNN, transformer)? Also, what if there are activation functions in the current used models (MLPs)? Have there been any preliminary tests or considerations for applying DASH to fields such as physics, material science, or other areas of computational biology?

### 4. Comparative Analysis
While DASH is compared with several pruning methods, are there any other recent state-of-the-art techniques or methods that should be included in the comparison? How does DASH stand against the very latest advancements in neural network sparsification?

**Limitations:**

The paper does discuss certain limitations, but more details could be added to enhance transparency and improve the assessment of the work:

**Scope of Application**: The generalizability of DASH to other domains outside gene regulatory networks isn't thoroughly explored. Adding more discussion on the potential limitations when applying DASH to different types of neural networks or scientific fields could be beneficial.

**Bias and Fairness**: The datasets used in biological research can sometimes contain biases, such as underrepresentation of certain populations. Discussing how the method handles such biases and ensuring that the models do not reinforce existing disparities is important. Suggesting ways to incorporate fairness checks and balance the representation in datasets would strengthen the ethical considerations.

---

> ### Author Rebuttal · Authors · 2024-08-07
>
> **Test set size**: For synthetic data, we know the *ground truth generative model* and evaluate on that. For breast cancer, we use 6\% of data to evaluate the MSE, which we picked as data is really scarce and we need sufficient number of samples to train the model. We do, however, have data of an *independent biological experiment* (the ChipSeq data) to evaluate the inferred model in terms of reconstructed GRN, hence the performance is also generalizable. For the yeast cell cycle data, we do test on an entire biological replicate (i.e., have one replicate for training, one for testing). We do see that this information might got lost in the details and will clarify in the revised manuscript.
>
> **Complexity of methodology**: The score is based on a basic reasoning that also underlies GRN inference techniques like [3], which argues that proxies of TF-TF interactions ($P$) or gene-gene interactions ($C$) are most readily available. The DASH pruning score aligns these proxies $P$ and $C$ with corresponding functions of the edges in the neural network. We have also provided more mathematical motivation below to Reviewer #4. We are happy to provide even more intuition on other aspects that the reviewer wants more clarity on.
>
> **Biological validation**: In terms of expert validation, one of our co-authors has deep expertise in molecular biology, and they have helped in the biological analysis of the GRNs inferred by DASH. We further provide the example of the following pathways: "... *TP53 activity* and *FOXO267 mediated cell death*, both of which are highly relevant in cancer [ 37 , 24 ]". Unfortunately, due to the strong space constraint, we had to present further biological validation in the appendix (see Section A). Furthermore, note that yeast cell cycle as well as breast cancer results were validated using ChIP-seq, which is an *independent experimental validation of biological plausibility*. As an additional result, we found an interesting biological insight, which we provide in the general rebuttal response due to character limit.
>
> **Numerical examples of $P$**: For synthetic data, we use a corrupted prior based on the data-generating model, which we elaborate on in App B.3 and B3.1. For real data, we use general information of transcription factor binding to gene promoter regions as prior information, which can be computed from binding motif matches with the corresponding genome (human respectively yeast). The result is a matching score that can be thresholded to get the {0,1} matrix encoding which (TF-encoding) gene has a relationship with which other gene. We follow the approach of Guebila et al. [1] to get matrix P. As prior C, we use the STRING database [2], which gives a general (i.e., not tissue-specific) graph of protein-protein interaction. Here, we use the interactions based on experimental evidence only and employ a cutoff of .6 to get a binary adjacency matrix. We will add the additional information to the discussion of priors in Appendix B.3.
>
> **Parameter tuning**: As mentioned in Appendix B4, the $\lambda$ values are determined using a $K$-fold cross validation approach. Specifically, we have automated a process that fits multiple models to the same dataset each with a different set of lambda values chosen from a grid. This automated grid-search is used to optimize the $\lambda$ values, based on predicted MSE on the validation set.
>
> **Generalizability**: We focused here on the considerably large field of gene regulatory networks, which has many applications in the biomedical domain. That said, we anticipate that DASH could be applicable to other core science domains, given the task's neural network structure follows a similar MLP layout and we have domain knowledge of similar structure (like matrices about input-output relationships $P$ and output-output relationships $C$).  In principle, the general definition of the score does not depend on the activation functions. The alignment of prior information with the pruning can be seen as an alignment of the prior with the amount of information that flows through the specific edges. Regarding CNNs and transformers, the application of DASH might be less straight-forward and requires further thinking on which part of the network (connections through filters, parts of attention heads) should be aligned with the prior. This is not directly evident, as such components are not directly interpretable in this context and are not necessarily associated with genes. We would be happy to add a detailed discussion about this aspect.
>
> **Comparative analysis**: We compared DASH to 8 other strategies for neural network sparsification, and additionally BioPrune and PINN+MP which are our own suggested strong baselines for comparison against DASH. Some of these eight strategies are classic benchmarks for comparison in the pruning literature, including IMP, which is considered as the standard baseline in the field, while others such as PathReg have been recently proposed for this exact problem setting. The PINN based method for inducing sparsity in the baseline PHOENIX model is as recent as 2024. We would appreciate specific references to relevant recent sparsification methods that include prior knowledge as we are not aware of any further ones.
>
> **Bias**: This is a great point, and we would be happy to discuss this as part of ethical considerations.
>
> [1] MB Guebila et al. GRAND: a database of gene regulatory network models across human conditions. Nucleic Acids Res. 2022
> [2] D Szklarczyk et al. The STRING database in 2023: protein-protein association networks and functional enrichment analyses for any sequenced genome of interest.  Nucleic Acids Res. 2023
> [3] D Weighill et al. Gene Regulatory Network Inference as Relaxed Graph Matching, AAAI, 2021

---

### Author Rebuttal · Authors · 2024-08-07

We sincerely thank the reviewers for their service and the provided constructive feedback. We are confident that we addressed the remaining concerns. In particular, we
- clarified the validation setting of GRNs, which stemmed from ChIP-seq experiments that were *independent* of the prior information,
- added additional results on a "simple" MLP architecture as base model (see rebuttal PDF),
- added an additional example of biological knowledge that can be extracted from our networks and sketch how it can be further used (see below),
- made mathematical notation consistent throughout the manuscript,
- added further explanations on the biological background, the relation of pruning scores and the application, and specifics on the training and validation process.

We report an additional finding on Heme signaling, which is a pathway uniquely identified as relevant in our approaches (cf. App. Fig. 5). Heme as a signaling molecule has key roles in the gene regulatory system [1], and turns out to have an anti-tumor role in breast cancer specifically [2]. Subsequent approaches pharmaceutically targeting Heme signaling showed success [3], with one of the key regulators affected being Bach1. To suggest further targets for, e.g., combination treatment, we hence examined the top-5 regulatory factors in terms of weights in our estimated gene regulatory dynamics. These factors include PBX1 and FOXM1, for which a drug repurposing of existing compounds, such as [4], could lead to a potential new treatment for this specific cancer.

[1] SM Mense, L Zhang, Heme: a versatile signaling molecule controlling the activities of diverse regulators ranging from transcription factors to MAP kinases.  Cell Res. 2006
[2] NA Gandini et al. Heme Oxygenase-1 Has an Antitumor Role in Breast Cancer.  Antioxid Redox Signal. 2019
[3] P Kaur et al. Activated heme synthesis regulates glycolysis and oxidative metabolism in breast and ovarian cancer cells. PlosOne 2021
[4] YA Shen et al. Development of small molecule inhibitors targeting PBX1 transcription signaling as a novel cancer therapeutic strategy. iScience 2021

---

### Decision · Program_Chairs · 2024-09-25

**Decision:**

Accept (poster)

**Comment:**

The scores from reviewers are 5, 5, 4, 3, 5. The reviews make many valuable points, but the authors have responded very well, so as the area chair, I believe that the numerical scores deserve to be higher, and this submission should be accepted.

The authors' responses include additional experiments with positive results, and a novel biological finding that is plausible and may have clinical utility.

Here is a high level question for the authors to discuss in the final version. They write "While pruning can simplify deep neural network architectures ... in the context of gene regulatory network inference, state-of-the-art techniques struggle with biologically meaningful structure learning." There are two different meanings for the word "network" here: the DNN versus the actual biological mechanisms. An ideal algorithm would output the actual regulatory structure, not just a pruned DNN that can answer questions about this. A DNN is an indirect representation of the actual mechanisms. The question is, to what extent can the literal regulatory structure be represented and discovered directly?

Another high-level suggestion: Mention in the abstract, and discuss more in the text, applicability of the DASH method outside molecular biology, expanding lines 61, 62 and 326, 327. In which domains does or does not prior knowledge tend to be of the type required by DASH?